# Examining the infographic design instructional process in terms of prospective mathematics teachers' infographic design proficiency, self-efficacy, and abilities in evaluating student errors: A model proposal

Neslihan Usta[1], Ali Özkaya[2]*, Gözdegül Arık Karamık[2], Yusuf Akın[1]

**1** Department of Mathematics and Science Education, Faculty of Education, Bartın University, Bartın, Türkiye, **2** Department of Mathematics and Science Education, Faculty of Education, Akdeniz University, Antalya, Türkiye

\* aliozkaya@akdeniz.edu.tr

## Abstract

The rapid integration of digital visualization tools into teacher education programs has created new opportunities for enhancing instructional design skills in mathematics education. This study investigates the impact of a technology-enhanced infographic design instructional process on prospective mathematics teachers' proficiency in digital infographic creation, their self-efficacy beliefs regarding technology-supported design and their abilities to evaluate middle school students' errors in the data processing learning domain. The proposed cyclical model integrates these dimensions into the digital infographic design process, utilizing cloud-based design platforms (e.g., Canva, Piktochart) to foster technological pedagogical content knowledge. The research was conducted with 45 prospective mathematics teachers enrolled in an elementary mathematics teacher education program at a public university in Türkiye's Mediterranean region during the 2021–2022 academic year. Data were collected through the Infographic Design Proficiency Rubric, the Infographic Design Self-Efficacy Scale, and pre- and post-tests on error evaluation. A single-group pre-test post-test quasi-experimental design was applied, supported by descriptive and content analyses of the digitally created infographics. Correlational analyses explored the relationships among the dependent variables. Findings revealed significant improvements in participants' technology-supported infographic design proficiency, self-efficacy perceptions, and error evaluation abilities. These results highlight the potential of integrating digital visualization tools into mathematics teacher education, and the study recommends their broader adoption to enhance instructional quality and learner engagement in STEM education.

**Data availability statement:** All relevant data are within the paper and its Supporting Information files and in the following external repository: https://doi.org/10.6084/m9.figshare.31283989.

**Funding:** The author(s) received no specific funding for this work.

**Competing interests:** The authors have declared that no competing interests exist.

## Introduction

Contemporary approaches to learning increasingly emphasize infographics, which are integrated visual artifacts that combine multiple representational elements such as charts, images, illustrations, and diagrams to communicate information in varied visual forms. Infographics stand out as tools that can facilitate learning and promote retention as they engage both visual and auditory channels when narrated or animated. Infographics, composed of words, graphics, and design elements, tell a story in a format that follows the introduction, main message, and conclusion parts [1, 2]. An infographic is a graphic representation of information that helps users visualize the "big picture" of an idea that might otherwise be difficult to understand [3]. According to Smiciklas [2], an infographic is a form of information graphic that blends data with design to convey concise messages from individuals or organizations to targeted audiences, and its primary components are graphics, words, and design features. The primary purpose of infographics is to convey an idea or information as a whole with the fewest possible graphics and words. Therefore, infographics aim for maximum information in minimum space [4]. The basic principles of infographic design include presenting information in a simple, clear, and visually appealing way and balanced use of design elements such as color, typography, and layout, and providing content suitable to the needs of the target audience [2]. A well-executed infographic should render complex data understandable and memorable by employing visual hierarchy and directional flow to support effortless processing [5]. Furthermore, data accuracy and source credibility must be ensured, and visual metaphors and icons should be leveraged judiciously to reinforce meaning [6]. A range of software environments can be used to create professional, high-impact infographics that support instructional goals, including *Adobe Illustrator, Canva, Piktochart, Tableau, and Venngage* [7].

### The importance of using infographics in education

Infographics constitute an engaging method for visually representing abstract concepts. Research shows that visual representations help students organize information more effectively and retain it for longer periods [8]. Creating an infographic is an engaging yet cognitively demanding assignment for students, requiring them to present the maximum amount of information within the constraints of time and space. Infographics also encourage active participation in learning through student-centered instructional approaches, supporting higher-order thinking skills such as analysis, synthesis, and evaluation [3, 9]. While preparing assignments with the aid of infographics, students can have the opportunity to develop general learning skills such as visual communication, collaboration, and critical thinking by attempting to express the subject visually [10]. This process enables learners to assume responsibility for their learning, reflect on core concepts, and communicate complex ideas clearly and creatively. Consequently, infographics can serve as an alternative assessment tool that promotes high-level achievement [10]. The way infographics present information helps learners more easily integrate new knowledge into existing schemata or use a pre-structured template to construct new ones [11].

By integrating text, images, graphics, and symbols to present information concisely and engagingly, infographics can serve as a bridge between visual learning and conceptual understanding in mathematics education, where students often struggle with abstract reasoning [12]. When mathematical information is visualized through infographics, it can help students better comprehend the relationships among concepts. Infographics are capable of visually depicting statistical data, geometric relationships, and mathematical applications, thereby fostering stronger connections to the subject matter. Mathematical content delivered via tools such as infographics enables learners to grasp knowledge in a visual and holistic manner. Through presentations enriched with graphs, icons, and color, students can perceive mathematical information more comprehensively and internalize it more effectively [13].

Designing rich learning environments is of paramount importance for enabling students to acquire a wide array of skills, including twenty-first century competencies. Enriching these environments hinges on teachers' and prospective teachers' ability to select, develop, and integrate visually rich instructional materials, in short, on their visual literacy skills [14]. Accordingly, prospective teachers must develop competence in visual literacy, enhance their lessons with visual elements, and become familiar with visualization methods and tools that can be employed in education so that they can cultivate their students' visual literacy skills. Infographics constitute a noteworthy example of such tools for visually representing information [15]. In mathematics instruction, two critical factors that directly influence teachers' or prospective teachers' professional competence and their students' academic achievement are the Pedagogical Content Knowledge (PCK) and Content Knowledge (CK) they possess [16]. While PCK underscores the importance of not only what is taught but also how it is taught, CK ensures that teachers have an accurate and in-depth understanding of mathematical concepts necessary for effective mathematics teaching [16]. CK enables teachers to establish relationships among mathematical concepts and to support students' inquiry and exploration processes [17]. It also helps them correct students' misconceptions and develop strategies for solving problems in multiple ways [18]. The integration of PCK and CK renders mathematics teaching more meaningful [19]. Therefore, to prepare prospective teachers for classroom practice and enable them to assess students effectively, instructional activities that allow them to understand and analyze students' thinking are required. In this context, it is essential to create various practice environments in which prospective teachers can identify and evaluate student errors. The ability to detect and assess student errors constitutes one of the fundamental building blocks of effective teaching [17], as it helps prospective teachers better understand students' learning processes and provide appropriate support. Most academic studies show that prospective teachers are generally successful at identifying student errors, although they may have deficiencies in certain areas [20, 21]. Furthermore, prospective teachers typically prefer methods such as asking open-ended questions, providing examples, and using visual materials when correcting student errors [17]. Some prospective teachers, however, tend to view student errors as personal failures [5]. Consequently, various methods can be explored to improve prospective teachers' abilities in effectively identifying and appropriately evaluating student errors, one of which is encouraging the use of information technologies such as infographics.

Data processing, one of the core strands of mathematics, is a learning domain dense with complex and abstract concepts. Consequently, visual tools are needed to teach and comprehend this area effectively. According to Smiciklas [2], using infographics in the data processing learning domain visually simplifies complex and voluminous data sets, making them more understandable and accessible. By emphasizing the relationships and trends within the data, information can be analyzed and interpreted rapidly [5]. For example, flowcharts can be employed to illustrate a stream of data, whereas bar charts or pie charts can facilitate comparisons among data points. Colorful, engaging, and interactive infographics can heighten students' interest in the topic and render the learning process more enjoyable [6]. The use of infographics in teaching the data processing learning domain is increasingly widespread, helping students grasp abstract data concepts and develop data literacy [13, 22, 23]. Alhadlaq [22], for instance, notes that infographics hold considerable potential to boost students' motivation and achievement when learning concepts in data processing and statistics. Because the data processing learning domain requires students to interpret graphs and conduct data analyses, it naturally intersects not

only with mathematics but also with other fields that depend on evidence-based argumentation, such as geography, and language arts [24].

## Significance of the study

In recent years, the number of studies focusing on infographics with higher education students has increased [25 - 26]. In addition, research has also been conducted with elementary school students [13] and with individuals who experience learning difficulties in mathematics [27, 28]. Interactive infographics have been reported to be an effective tool for enhancing second-grade elementary students' mathematics achievement [29]. In their meta-analysis, Elaldı and Çifçi [30] emphasized that the use of infographics exerts a positive and large effect on academic achievement, with the greatest effect size observed in the group exposed to four to five weeks of instruction and the smallest effect size in the group exposed to two to three weeks of instruction. In their scoping review, Jaleniauskiene and Kasperiuniene [31] noted that the use of infographics in higher education promotes active learning, assists students in creating diverse learning materials, and most frequently involves instructional practices in which ready-made infographics are used alongside student-generated ones.

Because infographics concretize and visualize abstract information, they stimulate students' curiosity about learning and play a facilitative role in the learning process [27]. In higher education, employing infographics has a significant effect on improving students' writing proficiency compared with traditional methods [32]. Likewise, by positively supporting higher education students' statistical literacy skills [33], studies have shown that infographics enhance learners' abilities to engage in statistical thinking and reasoning, to read and interpret data, to conclude journalistic texts, and to abstract data from statistical presentations. In a study by Salvatierra Melgar and colleagues [34], the importance of using mathematics infographics that incorporate geometric concepts as a visual tool in STEM learning was underscored. The intervention was found to bolster higher education students' creativity, critical thinking skills, and social interaction. Investigating concrete examples of climate mathematics through electronic posters, Sudakov et al. [35] emphasized that designing and using infographics facilitates learning in mathematics education and increases higher education students' active participation in class.

In the work of Kates and colleagues [36], students reported that the infographic design process helped them recall, review, and analyze information more thoroughly, and they stated that they intended to employ the process again. The infographic design experience improved students' comprehension, heightened their engagement, and, by fostering critical thinking skills, had a notable effect on knowledge retention. This finding highlights the advantages of providing higher education students with training in designing infographics [36].

A study conducted with elementary school students and teachers, in which an instructional design grounded in an infographic design model was developed, reported that participants held positive views concerning the use of infographics in learning environments [13]. In that study, teachers stated that infographics differ from other instructional materials because of their design structure, narrative quality, integration of numerous visual elements, and greater appeal, and they generally agreed that infographics can be employed in any type of lesson. The same study also examined teachers' self-efficacy perceptions regarding the design and use of infographics, revealing a statistically significant improvement after the intervention. Accordingly, teachers' self-efficacy increased with respect to designing infographics in digital environments, producing infographics that adhere to visual design principles and elements, and tailoring infographic design to the content and characteristics of the target audience. Similarly, Alwafi [25] investigated the effect of university students' self-created digital materials on deep learning approaches and self-efficacy. Conducted with a quasi-experimental design, the study found that by the end of the intervention, students in the experimental group had developed their deep learning approaches and self-efficacy to a greater extent than those in the control group. In another study, Wu and Kuwajima [26] reported that using infographics had positive effects on enhancing university students' English learning outcomes and motivation, meeting with high interest and motivation among the students. Examining prospective science teachers' perceptions of using self-designed infographics, Fadzil [37] found that, following the intervention, participants expressed positive opinions about

infographic assignments, participated actively in the learning process, and assumed responsibility by taking initiative in their learning. In this context, infographics represent suitable tools that can enhance the comprehension of conceptual knowledge by providing an engaging means of explaining scientific concepts. Designing an infographic requires deep reflection and compels creators to consider how best to convey fundamental ideas to the target audience [37].

The deep learning approach involves students' critical analysis of new ideas and their connection to previously understood concepts and principles [38, 39]. Whereas deep learning promotes genuine understanding, the surface learning approach entails students' passive acceptance of information and frequent attempts to memorize content without comprehension [40, 41]. Students who adopt a deep learning approach tend to perform better [25]. Self-efficacy, defined as a learner's belief in their capacity to complete a learning task successfully [42], influences performance, achievement, and the use of learning strategies [43]. Learners who hold strong beliefs in their abilities feel greater responsibility to design digital materials that motivate them to engage in learning activities [25]. Given these definitions, prospective teachers' ability to design infographics at an adequate level depends on the mental effort they devote to presenting the infographic to the target audience in the most effective manner, thereby necessitating substantial cognitive engagement [44]. Consequently, teachers and prospective teachers must possess the knowledge and skills required to integrate infographics into education alongside technological innovations [15,45,46]. The infographic design process strengthens thinking processes by requiring learners to research information, critically analyze its relevance, and condense content so that it fits within single-page templates [31]. Although a finished infographic presents condensed information, the selection process itself fosters critical thinking, opening the door to deep learning about what information to include and how best to present it. Acting as active constructors of their own knowledge encourages students to engage in more creative actions and to access richer learning opportunities. Evaluating infographics produced during the design process through peer and in-class discussions, and providing feedback, leads to improved learning outcomes [46]. Because time constraints can impede the development of visual thinking [47], additional classroom time should be allocated to encourage higher-order thinking and implement related activities [48]. The duration of exposure to data visualization determines what will be learned, interpreted, and experienced [1, 7, 23]. According to Nuhoğlu Kibar and Akkoyunlu [49], students should be given opportunities to create their own infographics through active learning, allowing them to become active rather than passive constructors of knowledge. The infographic design process prompts students to think critically about searching for resources and deciding what to include and how to summarize it concisely. This practice promotes meaningful learning, supports deeper information processing, and may spur more comprehensive research [50]. Infographics allow information to be created and represented in diverse formats, and the design process helps higher education students become more creative [51]. Likewise, teaching with infographics improves students' positive perceptions of the course and positively affects student interaction [52]. Research indicates that educators generally lack knowledge about how to use infographics in educational settings [31]. Insufficient integration of infographics into formal educational contexts makes it difficult for students to employ such communication tools effectively. Hence, understanding the purposes and methods for using infographics in higher education courses and observing the outcomes of exemplar implementations is essential. Evidence from the literature suggests that infographics can be employed effectively in teacher education. Within this context, the present study, using infographic design as an instructional method, aims to examine mathematics prospective teachers' levels of infographic design proficiency and infographic design self-efficacy, and to support their development of innovative strategies for analyzing student errors more effectively and offering corrective feedback. The widespread use of infographics in teacher education can enhance prospective teachers' visual literacy skills, thereby making learning processes more effective and enduring. This study seeks to contribute to the development of new approaches in both teacher education and mathematics instruction. By exploring the impact of the infographic design process on prospective teachers' abilities to identify and evaluate student errors in the data processing learning domain, the research underscores the importance of incorporating visual tools into mathematics education. Accordingly, the study proposes a cyclical model, depicted in Fig 1, to frame this instructional approach.

Fig 1 has been conceived as a cyclical model within the scope of the study. Accordingly, the practices carried out during the infographic design process exert a cyclical influence on enhancing prospective teachers' infographic design self-efficacy, raising their infographic design proficiency, and supporting their evaluation of student errors. Positive shifts in infographic design self-efficacy fortify prospective teachers' beliefs in their ability to teach the topic, thereby activating the infographic design process. Prospective teachers' favorable attitudes toward the content, combined with deep engagement, yield high-quality infographic products. Elevated belief and thorough engagement help prospective teachers identify student errors more readily and formulate corrective strategies. The intensive work inherent in the infographic design process strengthens prospective teachers' mastery of the subject matter, allowing them to use mathematical concepts, theories, and expressions accurately. By delineating the study's framework, Fig 1 underscores the importance of the research, as the cyclical model proposed through analysis of the core problem constitutes its principal outcome.

## Problem statement

The problem statement was determined as follows: "What is the effect of the infographic design process, employed as an instructional method in the data processing learning domain, on prospective mathematics teachers' infographic design proficiency, infographic design self-efficacy, and abilities in evaluating student errors?"

**Research questions.**

1. Is there a statistically significant difference between the scores obtained from the Infographic Design Proficiency Rubric before and after the intervention in the study group where the infographic design instructional method is used?

2. Is there a statistically significant difference between the scores obtained from the Infographic Design Self-Efficacy Scale before and after the intervention in the study group where the infographic design instructional method is used?

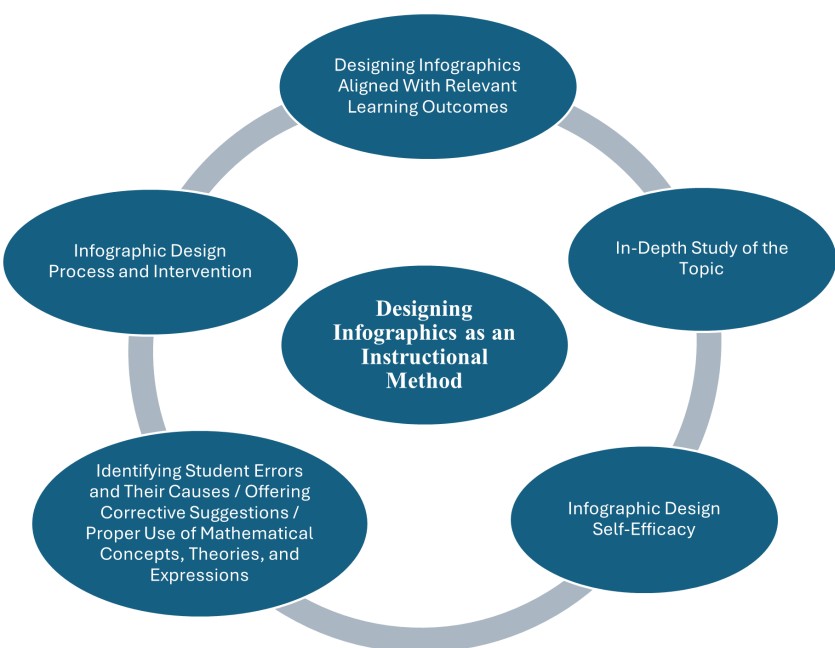

**Fig 1. Cyclical model manifested in the infographic design process.**

3. Is there a statistically significant difference between the scores obtained from the Error Evaluation Rubric, concerning middle school students' errors specific to the data processing learning domain, before and after the intervention in the study group where the infographic design instructional method is used?

4. Is there a statistically significant relationship among the scores that prospective teachers receive from the Infographic Design Self-Efficacy Scale, the Infographic Design Proficiency Rubric, and the Error Evaluation Rubric?

5. How does the change manifest before and after the intervention in prospective teachers' instructional explanations while evaluating middle school students' errors in the data processing learning domain in the study group where the infographic design instructional method is used?

## Method

### Research design

Because both qualitative and quantitative approaches were utilized in the study, a mixed methods design was adopted [53]. Although a single group was employed, and studies that rely on a single group are considered weak designs, the aim was to offset this weakness by reinforcing the study with qualitative findings. Incorporating qualitative data can strengthen a study by reducing the likelihood of issues such as internal validity threats and limited generalizability that may arise in single-group research [54, 55].

The independent variable of the study was the intervention implemented via the infographic design instructional method, whereas the dependent variables were the prospective teachers' infographic design proficiency, infographic design self-efficacy, and abilities in evaluating student errors. These dependent variables were investigated through both quantitative and qualitative procedures using rubrics and a scale. Qualitative data were analyzed through descriptive and content analyses, while quantitative data were processed with SPSS. In addition, correlational analysis was conducted to examine relationships among the dependent variables.

For the quantitative phase, a single–group pre–test–post–test quasi-experimental design was employed [56]. Prospective teachers' infographic design self-efficacy was measured with the Infographic Design Self-Efficacy Scale (IDSES). The infographics they produced were evaluated by three experts using the Infographic Design Proficiency Rubric (IDPR), and their assessments of student errors were scored by two experts using the Error Evaluation Rubric (EER). The qualitative component comprised analyses of the infographics designed by the participants according to the IDPR criteria and analyses of their error evaluation abilities according to the EER criteria. Fig 2 presents the overall research design.

### Research sample

The participants of this study comprised 45 prospective mathematics teachers enrolled in the third year of an Elementary Mathematics Teacher Education Program at a public university located in Türkiye's Mediterranean region. Information about the participants is presented in Table 1. Ethical approval was obtained prior to the study. All individuals who participated in the study were over 18 years of age. Participants were informed about the purpose and method of the research, the voluntary basis of participation, the protection of personal data, the use of the obtained data solely for scientific purposes, and their right to withdraw from the study at any time. Following this briefing, written informed consent was obtained from all participants. The consent forms were securely stored by the researcher. The participants took part in the study between September 2, 2022, and December 23, 2022. As a prerequisite, the prospective teachers had completed core courses such as Special Teaching Methods I–II, School Experience, Probability and Statistics, and Teaching Probability and Statistics. In this regard, they were assumed to possess adequate knowledge related to subject matter pedagogy. To ensure confidentiality, real names were not used, and the participants were assigned codes such as PT1, …, PTn.

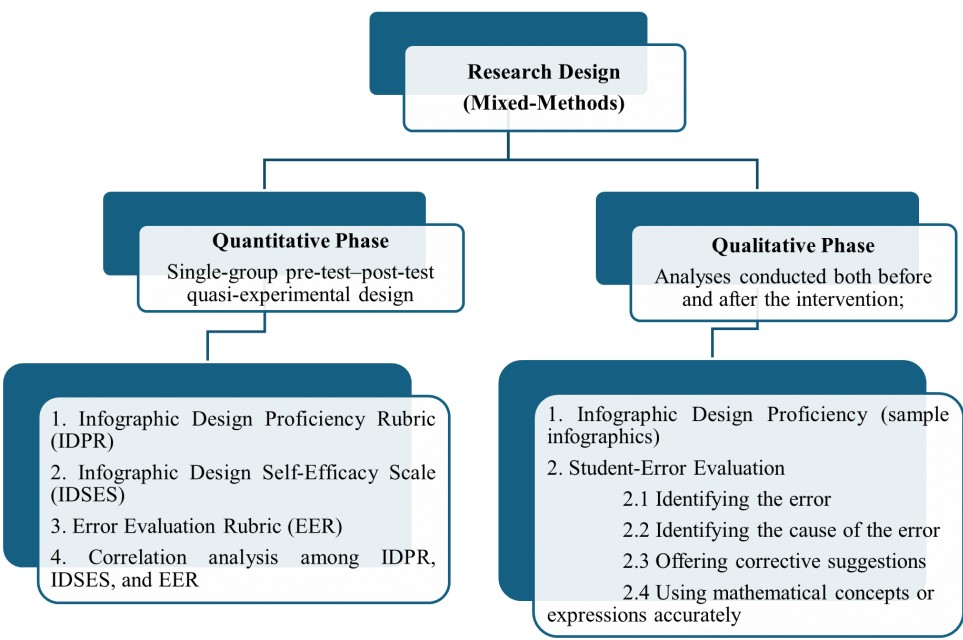

**Fig 2. Research design and methods employed.**

**Table 1. Participant information.**

| Participants | Frequency | Percent (%) |
|---|---|---|
| Male students | 15 | 33.3 |
| Female students | 30 | 66.7 |

## Ethical considerations

This study was conducted in accordance with the ethical principles governing research involving human participants in educational settings. After obtaining permission from the developers of the measurement instruments, the overall study protocol was reviewed and approved by the Akdeniz University Social and Human Sciences Scientific Research and Publication Ethics Committee (decision no.08/163 dated 20.04.2022). Additional authorization was obtained from the relevant faculty of education prior to data collection. Participation was voluntary, and informed consent was obtained from all participants before the study was conducted.

## Data collection instruments

**Infographic design proficiency rubric (IDPR).** The purpose of the IDPR is to evaluate prospective teachers' infographic designs against specific criteria both before and after the intervention. The IDPR was initially developed by Davidson [57] and Lamb & Johnson [3] and later refined by Turan Güntepe and Dönmez Usta [58]. The rubric primarily focuses on balancing functionality (knowledge transfer) and aesthetics (visual appeal) in infographic design, together with the appropriate use of educational objectives. The infographics created by prospective teachers were assessed according to the categories "adequate," "partially adequate," and "inadequate" as defined in the IDPR. In the present study, three experts reviewed the participants' infographics. For inter-rater reliability, "adequate" responses were assigned 2 points, "partially adequate" responses 1 point, and "inadequate" responses 0 points. As shown in Table 2, the IDPR comprises

**Table 2. Infographic design proficiency rubric (IDPR).**

| Criteria | Categories | | |
|---|---|---|---|
| | Adequate (2 points) | Partially Adequate (1 points) | Inadequate (0 points) |
| 1. Creating simple information graphics | | | |
| 2. Presenting complex information quickly and clearly | | | |
| 3. Presenting information through visuals and text | | | |
| 4. Using a limited number of words and text in information graphics | | | |
| 5. Including explanatory text | | | |
| 6. Designing an appealing and attractive layout for the reader | | | |
| 7. Preparing in accordance with design principles | | | |

Source: [58].

seven criteria, each with three possible ratings. The minimum attainable score is 0, and the maximum is 14. When a prospective teacher fully and correctly meets a criterion, the work is rated as adequate (2 points). Partial fulfillment yields a partially adequate rating (1 point). Non-fulfillment results in an inadequate rating (0 points).

**Infographic design self-efficacy scale (IDSES).** The self-efficacy perception scale for infographic design, developed by Özdamlı and Özdal [13], comprises 35 items distributed across three factors. To establish reliability for the present study, the scale was administered to 262 prospective teachers. An examination of the data indicated approximate normality (Skewness (-,179), Kurtosis (-,563)). The Cronbach's Alpha for the first sub-factor of the scale, "competencies related to the design of infographics in a digital environment" was .954, the second sub-factor, "competencies related to designing infographics according to visual design principles" had a Cronbach's Alpha of .932, the third sub-factor, "competencies related to designing infographics according to content and target audience appropriateness" had a Cronbach's Alpha of .978, and the overall Cronbach's Alpha was .981. The full Infographic Design Self-Efficacy Scale is provided in Appendix 1.

**Error evaluation rubric (EER). Error evaluation pre-test (EEPrT) and error evaluation post-test (EEPoT):** The study employed an EEPrT and an EEPoT to have prospective mathematics teachers evaluate middle school students' errors specific to the data processing learning domain and to propose remedies for those errors. Each test consists of items that include students' incorrect or erroneous responses drawn from the data processing domain. Specifically, both instruments contain five open-ended questions adapted from validated and reliable sources in the literature [59–61] that elicit a variety of typical student errors. When selecting questions, care was taken to ensure that (a) the errors committed by students were explicit and (b) the items represented different error types. Prospective teachers were first asked to determine whether the student responses contained an error. If so, they identified the error and offered corrective strategies. Sufficient time was provided to complete the tasks. Participants were instructed to answer each question in line with Gökkurt's [62] guidelines. Guidelines are as follows. a) Based on the student's response, do you think the student made an error? If so, what is the error? What might be the possible reason(s) for this error? b) What question(s) could you pose to help the student recognize the error? c) Which key mathematical concept or prerequisite knowledge could you employ to help the student arrive at the correct answer? What instructional methods, techniques, or strategies could you use to address the error?

The data obtained from the EEPrT and EEPoT were analyzed using the EER, which incorporates the categories and subcategories proposed by Gökkurt and colleagues [63] along with additional categories introduced by the present researchers. Table 3 lists the relevant learning outcomes in data processing, associated concepts and terms, and the item numbers and titles used in the tests. Appendix 2 provides sample items from the open-ended EEPrT and EEPoT instruments.

**Table 3. Data processing learning domain learning outcomes, related concepts/terms, and alignment of EEPrT/EEPoT items.**

| Learning Outcomes | Related Concepts/Terms | Test item number and title | |
|---|---|---|---|
| | | **Pre-Test** | **Post-Test** |
| Formulates research questions that require data collection and collects data relevant to those questions. | Research question, sampling, data collection and interpretation | Question 1 Muğla Vacation | Question 1 Charity Campaign |
| Interprets a pie chart for at least one data set. | Pie chart | Question 2 Ice Age 5 Movie | Question 2 Vote Distribution |
| Uses arithmetic mean and range to compare and interpret data from two groups. | Minimum and maximum values, range, arithmetic mean, mode, median | Question 3 Football Match Game on Computer | Question 3 Umut–Sevgi Middle Schools |
| Represents data with bar, pie, or line graphs and converts appropriately among these representations. Interprets line and bar graphs involving up to three data sets. | Line, pie, and bar graphs. Data interpretation | Question 4 Student Counts Table | Question 4 Foreign Language Instruction |
| Constructs and interprets a line graph. Determines and interprets the mean, median, and mode for a data set. | Line graph, data set, mode, median, mean. | Question 5 Number of Train Services | Question 5 Atakent–Pelitköy High School |

The categories and subcategories of the EER, which are developed to analyze the responses that the prospective teachers gave to the items in the EEPrT and EEPoT, are presented in Table 4. The subcategories of each category are assigned with a scoring scheme of 3-2-1-0.

In Table 4, the subcategories under the *Identifying the error* category are explained as follows, as an example. *Unable to identify the error* refers to situations in which prospective teachers cannot identify the error or provide no response. *Incorrect identification of the error* denotes cases in which their answers are entirely incorrect. *Partially correct identification of the error* describes responses that do not fully address the question's requirements, answers that contain minor mistakes, or only a small amount of correct information. *Correct identification of the error* indicates responses that are completely accurate and meet the expected criteria.

## Intervention process

The infographic design instruction and implementation of the research were completed in 14 weeks, and data collection was finalized in 2 weeks. In the 16-week study, EEPrT and IDSES were administered to the prospective teachers as

**Table 4. Error evaluation rubric (EER).**

| 1 Identifying the error | 2 Identifying the cause of the error | 3 Offering a corrective suggestion | 4 Use of mathematical concept / expression |
|---|---|---|---|
| 1a: Unable to identify the error / no response | 2a: Unable to identify cause of the error / no response (0 points) | 3a: No solution proposed (0 points) | 4a: No mathematical concept or expression used (0 points) |
| 1b: Incorrect identification of the error (1 points) | 2b: Incorrect cause of the error / wrong explanation (1 points) | 3b: Incorrect solution proposed (1 points) | 4b: Only one mathematical concept or expression used (1 points) |
| 1c: Partially correct identification of the error (2 points) | 2c: Partially correct cause of the error (2 points) | 3c: Partially correct solution proposed (2 points) | 4c: At least two mathematical concepts or expressions used (2 points) |
| 1d: Correct identification of the error (3 points) | 2d: Correct cause of the error / Correct explanation (3 points) | 3d: Correct solution proposed (3 points) | |

pre-tests. Following the pre-tests, general information about infographics was presented for four weeks ($4 \times 3 = 12$ class hours). These presentations covered the definition of an infographic, purposes for using infographics, points to consider when preparing them, and an introduction to design software (e.g., Canva, Visme, Venngage, Piktochart, Mind the Graph), while sample infographics were examined. At the end of the presentations, the prospective teachers were asked to design various infographics freely, and one week was allotted for this task. The designs were evaluated in terms of purpose, adherence to visual design principles, suitability for the target audience, and the appropriateness of visually and textually presenting complex information. The results were discussed in class with the aid of an interactive whiteboard, incorporating the prospective teachers' views. Owing to its ease of use, drag-and-drop interface, and free platform, the prospective teachers preferred Canva. In conjunction with this activity, they revisited the infographics they had independently designed and made necessary revisions. This stage of the intervention was completed in two weeks ($2 \times 3 = 6$ class hours).

Immediately after this preliminary activity targeting the infographic design process, the prospective teachers were asked to design various infographics related to the data processing learning domain. These infographics constituted the pre-test data collected to address the sub-problem concerning the determination of the prospective teachers' infographic design proficiency. The pre-test data were evaluated using the IDPR. Following the preliminary activity and the collection of the pre-test data, the main intervention, which is namely, the infographic design instructional process as an instructional method, was initiated. During the main intervention, the dimensions of infographic design principles, identifying the target audience, gathering relevant information, selecting the infographic type appropriate to the purpose, constructing a logical hierarchy, choosing an appropriate template for designing an infographic, and customizing that template were explained in detail through illustrative examples. Throughout the eight-week (24-class-hour) main intervention, the prospective teachers engaged in in-depth work and designed three infographics each within the data processing learning domain. Each infographic was discussed in class with the aid of an interactive whiteboard according to the IDPR criteria. The third infographic served as the post-test data and was evaluated using the IDPR. At the end of the main intervention, EEPoT and IDSES were administered to the prospective teachers as post-tests for a second time. Table 5 presents the implementation process of the study, the procedures conducted, and the purposes of those procedures.

## Data analysis

IDPR was employed to determine the level of quality of the infographics that the prospective teachers designed for the data processing learning domain. Three experts independently evaluated the infographics. Inter-rater agreement was examined using the kappa coefficient [64]. Because concordance among three raters was assessed, Fleiss' kappa was calculated, yielding a value of .82. Using the interpretive guidelines proposed by Landis and Koch [65], this coefficient indicated a strong level of agreement among raters. In selecting the appropriate statistical procedures, the normality of the variables was evaluated [56]. The distribution of the dataset that met the majority of normality criteria was treated as normal, and parametric techniques were applied. After confirming consistency among raters' scores on the IDPR, the arithmetic mean of the three experts' ratings was computed. Given the normal distribution of the data, a paired samples t-test was conducted to determine whether a significant difference existed between the prospective teachers' pre and post intervention IDPR scores.

The prospective teachers' self-efficacy perceptions regarding infographic design were evaluated with the IDSES before and after the intervention, and a Shapiro–Wilk test was applied to determine whether the data followed a normal distribution. According to the Shapiro–Wilk results, the variables exhibited normality for both the pre-test data (Statistic = .971, p = .326) and the post-test data (Statistic = .954, p = .070). Consequently, a paired-samples t-test was used for comparison.

The prospective teachers' responses to the open-ended items in the EEPrT and EEPoT were analyzed with the EER. Drawing on the categories and subcategories proposed by Gökkurt et al. [63], a descriptive analysis was conducted. Additional categories and subcategories devised by the present researchers (e.g., 4. Use of mathematical concepts/expressions 4a, 4b, 4c) were then incorporated through content analysis. The resulting categories constitute the rubric's

Table 5. Procedures performed during the intervention phase of the study and their purposes.

**Intervention Phase**
**(14 weeks + 2 weeks; 14 × 3 = 42 class hours, 2 × 3 = 6 class hours)**

|  | Procedure | Purpose of the Procedure |
|---|---|---|
| Pre-Intervention | Administered IDSES and EEPrT | Obtaining IDSES pre-test and EEPrT data |
| Preliminary phase | 1. Provided a PowerPoint presentation covering general information on infographics and the design process.<br>2. Introduced infographic design software (e.g., Canva, Piktochart).<br>3. Asked prospective teachers to freely design infographics on various topics.<br>4. Displayed the infographics on an interactive whiteboard and evaluated them using IDPR.<br>5. Requested prospective teachers to design infographics specific to the data processing learning domain. These infographics, evaluated with IDPR, constituted part of the qualitative data and, through scoring, part of the quantitative pre-test data. | Familiarizing prospective teachers with infographics, the design process, and relevant software.<br>Experiencing the process by designing an infographic on a free topic.<br>Using the infographics designed related to data processing learning domain as qualitative and quantitative pre-test data. |
| Main Intervention | 1. Provided instruction on infographic design.<br>2. Prospective teachers used Canva to design infographics specific to the data processing learning domain.<br>3. Conducted in-depth design work.<br>4. Each prospective teacher created three infographics.<br>5. Each infographic was discussed in class via an interactive whiteboard according to IDPR criteria. The third infographic served as post-test data. | Designing infographics specific to data processing learning domain.<br>Engaging in in-depth study.<br>Correcting incomplete or incorrect content by focusing on relevant mathematical concepts and expressions.<br>Using the infographics designed related to data processing learning domain as qualitative and quantitative post-test data. |
| Post Intervention | 1. Administered IDSES and EEPoT. | Obtaining IDSES post-test and EEPoT data. |

criteria: Identifying the error (IE), Identifying the cause of the error (ICE), Proposing a solution (PS), and using appropriate mathematical concepts/expressions (UMC). The EEPrT and EEPoT were coded independently by two researchers according to the EER criteria, after which consensus scores were established. Using Miles and Huberman's [66] formula, an intercoder agreement rate of 93% was obtained. Discrepancies were resolved through discussion until full agreement was reached. Scores derived from the EER's categories and subcategories were transferred to SPSS and analyzed quantitatively. Because the EER data did not follow a normal distribution, the non-parametric Wilcoxon signed-rank test was employed [56]. In addition, following Cohen's [67] small–medium–large classification, the effect size coefficient was calculated with the formula $r = \frac{Z}{\sqrt{n}}$ [68,69]. Effect sizes for the r coefficient were classified as small for values below 0.30, medium for values between 0.30 and 0.50, and large for values of 0.50 and above.

Correlation analysis was conducted to determine whether statistically significant relationships existed among the scores obtained from IDPR, IDSES, and EER. A correlation coefficient between 0.00 and 0.25 indicates a very weak relationship, a value between 0.26 and 0.49 denotes a weak relationship, and a value between 0.50 and 0.69 reflects a moderate relationship [70].

Sample infographics designed by the prospective teachers were examined using descriptive analysis in line with the IDPR criteria. In addition, the prospective teachers' instructional explanations of middle school students' errors specific to the data processing learning domain, elicited through the EEPrT and EEPoT, were assessed via content analysis. Changes in these explanations from pre-intervention to post intervention were then examined, and their frequency counts are presented in tabular form.

## Results

### Quantitative results

**Findings and interpretation for the first sub-problem.** The infographics designed by the prospective teachers for the data processing learning domain were evaluated with IDPR. A paired samples t-test was conducted to determine

whether the mean scores obtained before and after the intervention differed significantly. The results are presented in Table 6.

Examination of Table 6 shows a statistically significant difference between the prospective teachers' pre-intervention ($X=7{,}07$, sd$=0{,}80$) and post-intervention ($X=10{,}13$, sd$=0{,}72$) IDPR mean scores. The paired samples t-test yielded t$=-18.63$, and this difference was found statistically significant (p$<.001$). This result shows that the infographic design instructional method effectively enhances prospective teachers' infographic design proficiency. Presenting the infographic design process systematically as an instructional method substantially improved the prospective teachers' abilities in information visualization, content organization, and audience-appropriate design.

**Findings and interpretation for the second sub-problem.** A paired samples t-test was carried out to examine whether a statistically significant difference existed between the scores that the prospective teachers obtained from IDSES before and after the intervention. The results are summarized in Table 7. As Table 7 shows, a significant difference in favor of the post-test emerged across every subfactor and for the overall scale score.

Examination of Table 7 reveals that the infographic design instructional intervention produced a statistically significant effect across all factors [($t_{factor1}=-6{,}71$; p$<0{,}001$), ($t_{factor2}=-8{,}13$; p$<0{,}001$), ($t_{factor3}=-11{,}65$; p$<0{,}001$), ($t_{total}=-11{,}06$; p$<0{,}001$)]. The greatest change was observed in the subfactor "competence in designing infographics appropriate to content and target audience" (d$=2.31$). In addition, the effect size for the total score was likewise very large (d$=2.23$). These findings indicate that the applied method is highly effective in enhancing prospective teachers' self-efficacy for infographic design.

**Findings and interpretation for the third sub-problem.** To examine whether the infographic design instructional method produced a statistically significant difference in prospective teachers' ability to evaluate middle school students' errors specific to the data processing learning domain, scores obtained from the EEPrT and EEPoT were compared with the Wilcoxon signed-rank test. Means, standard deviations, and Shapiro–Wilk normality values for the study group are presented in Table 8.

**Table 6. Paired samples test of IDPR pre- and post-test scores.**

| | | N | X | sd | df | t | p |
|---|---|---|---|---|---|---|---|
| IDPR | Pre-test | 45 | 7,07 | 0,80 | 44 | −18,63 | <0,001 |
| | Post-test | 45 | 10,13 | 0,72 | | | |

**Table 7. Paired samples t-test results for IDSES pre- and post-test scores.**

| Factor | Test | N | X | sd | t | p | d |
|---|---|---|---|---|---|---|---|
| Factor 1* | Pre-test | 45 | 34,49 | 11,16 | −6.71 | <0,001 | 1,35 |
| | Post-test | 45 | 47,20 | 4,80 | | | |
| Factor 2** | Pre-test | 45 | 22,20 | 10,42 | −8.12 | <0,001 | 1,50 |
| | Post-test | 45 | 35,96 | 5,13 | | | |
| Factor 3*** | Pre-test | 45 | 36,20 | 13,60 | −11.65 | <0,001 | 2,31 |
| | Post-test | 45 | 62,84 | 6,79 | | | |
| TOTAL | Pre-test | 45 | 92,89 | 28,15 | −11.06 | <0,001 | 2,23 |
| | Post-test | 45 | 146,00 | 14,91 | | | |

(p<0.001),

*: Competence in designing infographics within digital environments.

**: Competence in designing infographics in accordance with visual design principles.

***: Competence in designing infographics appropriate to content and target audience.

**Table 8. Descriptive statistics for EEPrT and EEPoT scores.**

| EER Criterion | N | Pre-Test | | | Post-Test | | |
|---|---|---|---|---|---|---|---|
| | | $\overline{X}$ | sd | Shapiro-Wilk (p) | $\overline{X}$ | sd | Shapiro-Wilk (p) |
| IE – Identifying the Error | 45 | 11.44 | 2.599 | .005 | 13.38 | 2.003 | .000 |
| ICE – Identifying the Cause of the Error | 45 | 9.69 | 2.466 | .008 | 12.44 | 2.272 | .001 |
| PS – Proposing a Solution | 45 | 6.98 | 2.856 | .536 | 9.58 | 3.306 | .119 |
| UMC – Using Mathematical Concepts/Expressions | 45 | 4.84 | 2.163 | .170 | 6.62 | 2.124 | .048 |
| TOTAL | 45 | 32.96 | 9.247 | .326 | 42.02 | 8.532 | .070 |

Table 8 shows that the prospective teachers' post-test scores increased across all criteria after the infographic design intervention. For the IE and ICE criteria, the Shapiro–Wilk p values for both pre-test and post-test are below .05, indicating non-normal distributions. Therefore, non-parametric analysis was required. Accordingly, the Wilcoxon signed-rank test was employed to assess statistical differences for these criteria, and the results are reported in Table 9.

Examination of Table 9 reveals significant differences between the pre- and post-test scores for IE, ICE, PS, UMC, and the overall error evaluation total. Specifically, the infographic design instructional method produced statistically significant improvements in prospective teachers' ability to identify student errors (IE; $z = -4.971$, $p < .05$), identify the causes of those errors (ICE; $z = -3.717$, $p < .05$), propose corrective solutions (PS; $z = -4.331$, $p < .05$), and use appropriate mathematical concepts and expressions (UMC; $z = -4.200$, $p < .05$), as well as in their overall error evaluation performance ($z = -4.850$, $p < .05$). Table 9 further indicates that all observed differences are associated with large effect sizes. The intervention accounts for approximately %30 of the variance in IE, %54 in ICE, %41 in PS, %39 in UMC, and %52 in the total error evaluation score.

**Findings and interpretation for the fourth sub-problem.** Whether statistically significant relationships exist among the scores that the prospective teachers obtained from the IDSES, IDPR, and EER (EEPrT and EEPoT) was investigated.

**Table 9. Wilcoxon signed-rank test results for EEPrT and EEPoT scores.**

| EER-Criteria | Posttest-Pretest | N | Mean Rank | Rank Sum | z | p | Effect Size |
|---|---|---|---|---|---|---|---|
| IE | Negative Rank | 7 | 11.64 | 81.50 | −3.717* | .000 | r=0.55, r²=0.30 Large effect |
| | Positive Rank | 27 | 19.02 | 513.50 | | | |
| | Equal | 11 | – | – | | | |
| ICE | Negative Rank | 5 | 12.70 | 63.50 | −4.971* | .000 | r=0.74, r²=0.54 Large effect |
| | Positive Rank | 38 | 23.22 | 882.50 | | | |
| | Equal | 2 | – | – | | | |
| PS | Negative Rank | 7 | 13.93 | 97.50 | −4.331* | .000 | r=0.64, r²=0.41 Large effect |
| | Positive Rank | 34 | 22.46 | 763.50 | | | |
| | Equal | 4 | 13.07 | 91.50 | | | |
| UMC | Negative Rank | 7 | 21.52 | 688.50 | −4.200* | .000 | r=0.62, r²=0.39 Large effect |
| | Positive Rank | 32 | – | – | | | |
| | Equal | 6 | | | | | |
| TOTAL | Negative Rank | 6 | 14.67 | 88.00 | −4.850* | .000 | r=0.72, r²=0.52 Large effect |
| | Positive Rank | 39 | 24.28 | 947.00 | | | |
| | Equal | 0 | – | – | | | |

*: Based on negative ranks.

Pearson correlation (r) test results for the pre-test and post-test scores of the IDSES, IDPR, and EEPrT–EEPoT are presented in Table 10.

Examination of Table 10 shows a weak yet statistically significant positive correlation between the EEPrT scores and the pre-test IDPR scores of the prospective teachers (r = .385; p = .01). This finding indicates that, prior to the intervention, a relationship existed between the teachers' infographic design proficiency and their ability to evaluate middle school students' errors. In other words, prospective teachers with higher design proficiency tended to perform better in assessing students' errors data processing. However, no significant correlations were found among the other instruments, particularly between the IDSES and either the IDPR or the EEPrT/EEPoT scores (p > .05). This suggests that the teachers' self-efficacy perceptions regarding the infographic design process may not be directly linked to their actual design performance (IDPR) or to their error-evaluation abilities (EEPrT/EEPoT). In the analysis of the post-test scores, no statistically significant relationships emerged among any of the measures (p > .05). This result implies that, following the intervention, improvements on each scale occurred independently, indicating that the instructional program may have influenced the various ability domains separately.

## Findings from the qualitative analysis of the data

To enrich the interpretation of the quantitative results, a qualitative analysis of the data was conducted. Initially, to determine the prospective teachers' levels of infographic design, samples of the static infographics they produced for the data processing learning domain before and after the intervention were analyzed qualitatively according to the IDPR. Secondly, the instructional explanations that the prospective teachers provided in the EEPrT and EEPoT regarding middle school students' errors specific to the data processing domain were examined using qualitative methods. Changes in these explanations were investigated, and the frequencies of the resulting categories are presented in tabular form. The tables are supported with excerpts from the participants' responses.

**Findings and interpretation from evaluating the prospective teachers' infographics with the IDPR criteria.** The prospective teachers created their infographics using Canva. After the software introductions, they were asked which platform they preferred. The vast majority chose Canva. Their explanations were as follows. *"This platform is free and easier to use than the others. Thanks to its drag-and-drop interface, we don't need much technical knowledge. We can design with ease. It offers ready-made templates, and we can use them for many different topics. We can enrich our infographics with free visuals, icons, and graphic elements. If we don't like something or want to change a part of the infographic, we can collaborate with our peers, revise it together, give each other feedback, and share our presentations."* Figs 3-6 present examples of the static infographics the prospective teachers produced on freely chosen topics during the preliminary phase. In these examples, the participants designed infographics introducing logic and strategy games such as tangram, mangala, and chess.

**Table 10. Pearson correlations (r) among IDSES, IDPR, and EER scores (pre-test and post-test).**

| Test | Scale/ Factor | Pearson Correlation | 1 | 2 | 3 |
|------|---------------|---------------------|---|---|---|
| Pre-test | IDSES (1) | | -- | | |
| | IDPR (2) | r | −0.136 | -- | |
| | EEPrT (3) | | 0.171 | 0.385** | -- |
| Post-test | IDSES (1) | | -- | | |
| | IDPR (2) | r | 0.183 | -- | |
| | EEPoT (3) | | 0.084 | 0.166 | -- |

** Correlation is significant at the p=0.01 level.

Fig 3 presents an infographic in which a prospective mathematics teacher attempted to present the definition of the tangram game, its historical origin, and the information that it consists of seven geometric pieces. The prospective teacher also attempted to express that the process of playing the game, which is based on forming a specific shape, is divided into easy, medium, and difficult levels. In addition, the prospective teacher explained the basic rules of the game, such as using all pieces, not overlapping the pieces, and placing them so that they touch each other.

Fig 4 presents an infographic in which a prospective mathematics teacher attempted to show that the Mangala game is played by two players, that there are 12 small pits and two stores on the game board, and that the game is played with 48 stones. The prospective teacher expressed that players distribute the stones into the pits in order to accumulate the highest number of stones in their own store and that the game is won by the player who collects the most stones. In addition, the prospective teacher attempted to explain four basic rules of the game, the distribution of the stones, the interaction with the opponent's pits, and the conditions for ending the game with the support of example visuals.

Both the Tangram and Mangala designs feature a title and a summary. Each of them also includes a QR code that enables readers to watch videos, learn about the games, and play them. A reference section identifying the information source and providing the relevant website stands out as well. In the Tangram infographic, color harmony has been observed. In both infographics, the information is legible and presented in logically organized subsections. However, in the section intended to convey the key message, visual elements should have been used more effectively, and explanations offered with a limited number of words. Instead, an excessive amount of text, neither memorable nor appealing to readers, predominates.

Fig 5 presents an infographic in which a prospective teacher attempted to indicate that the chess game is played between two players on an 8x8 board with 32 pieces and that a total of 64 squares are arranged in black and white. The prospective teacher expressed that each player has 16 pieces at the beginning of the game, the types of these pieces, and that the aim of the game is to checkmate the opponent's king. In addition, the prospective teacher attempted to explain how the chess pieces (king, queen, rook, bishop, knight, and pawn) move and their basic functions.

Fig 6 presents an infographic in which a prospective teacher expressed that chess is played by two players on an 8 × 8 board, that the aim of the game is to checkmate the opponent, and that it is an effective game for developing intelligence. The prospective teacher attempted to explain how many pieces there are (pawn, rook, knight, bishop, queen, and king) and how they move with the support of visuals. In addition, the prospective teacher added to the text that approximately 600 million people worldwide know how to play chess.

Similarly, the chess infographics shown in Figs 5 and 6 bear titles and are legible. However, neither infographic includes a reference list, nor does either offer features likely to capture and sustain readers' interest or prompt them to take action, such as embedding a QR code or providing a website address. In the infographic designed as black and white, the heavy reliance on text makes it difficult for readers to grasp the intended key message, and the haphazard placement of text blocks distracts attention. In the infographic designed with color, by contrast, violates basic visual design principles. Numerous colors are employed without regard for harmony, the main message is not emphasized, and no attention is paid to hierarchy, proportion, or rhythm among the informational elements.

On the whole, findings derived from evaluating the infographics created on free topics during the preliminary phase according to the IDPR criteria can be summarized as follows. These infographics generally present up-to-date and accurate information, cite their sources, and offer readers various opportunities for further engagement by including QR codes or website addresses. On the other hand, they typically contain excessive amounts of text, lack compelling designs capable of attracting and maintaining interest, fail to highlight a core message, present information without memorable structure or proportion, scatter text randomly across the canvas, leave relationships among text elements underdeveloped, and largely disregard design principles. After designing infographics on free topics, the prospective teachers proceeded at the start of the main intervention to create infographics specific to the data processing learning domain. Figs 7 and 8 present examples of those infographics produced during the main phase of the study.

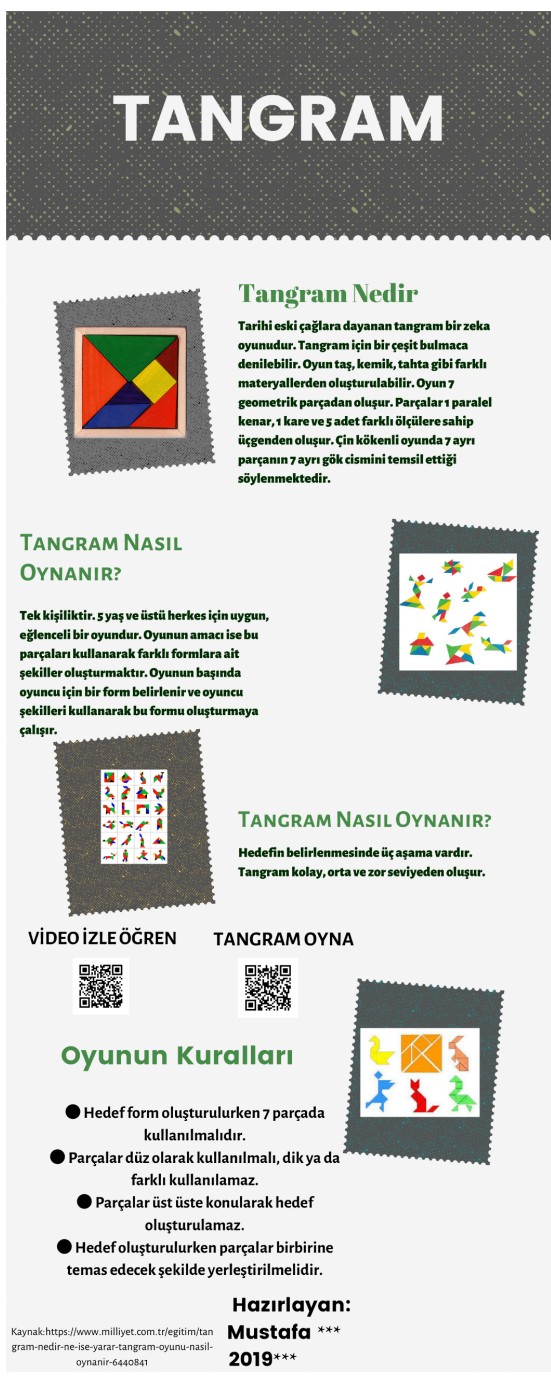

**Fig 3. Infographic on Tangram designed on a free topic during the preliminary phase.**

Fig 7 presents an infographic in which a prospective teacher presented numerical information regarding wheat waste, the monetary value of bread, daily and annual consumption, and waste amounts related to bread waste. The prospective teacher also showed frequency tables and graphical distributions regarding how stale bread is used by presenting the

# Mangala

**TÜRK ZEKA VE STRATEJİ OYUNU**

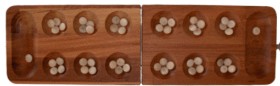

Mangala Türk Zeka ve Strateji Oyunu iki kişi ile oynanır. Oyun tahtası üzerinde karşılıklı 6'şar adet olmak üzere 12 küçük kuyu ve her oyuncunun taşlarını toplayacağı birer büyük hazine bulunmaktadır. Mangala Oyunu 48 taş ile oynanır.

Oyuncular 48 taşı her bir kuyuya 4'er adet olmak üzere dağıtırlar. Oyunda her oyuncunun önünde bulunan yan yana 6 küçük kuyu, o oyuncunun bölgesidir. Karşısında bulunan 6 küçük kuyu rakibinin bölgesidir. Oyuncular haznelerinde en fazla taşı biriktirmeye çalışırlar. Oyun sonunda en çok taşı toplayan oyuncu oyun setini kazanmış olur. Oyuna kura ile başlanır. Oyunda 4 ana temel kural vardır.

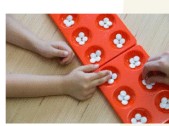

**1. TEMEL KURAL :** Kura neticesinde başlama hakkı kazanan oyuncu kendi bölgesinde bulunan istediği kuyudan 4 adet taşı alır. Br adet taşı aldığı kuyuya bırakıp saatin ters yönünde, yani sağa doğru her br kuyuya birer adet taş bırakarak elindeki taşlar bitene kadar dağıtır. Elindeki son taş hazinesine denk gelirse, oyuncu tekrar oynama hakkına sahip olur. Oyuncunun kuyusunda tek taş varsa, sırası geldiğinde bu taşı sağındak kuyuya taşıyabilir. Hamle sırası rakibine geçer. Her seferinde oyuncunun elinde kalan son taş oyunun kaderin belirler.

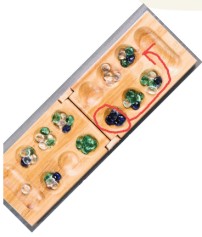

**2. TEMEL KURAL:** Hamle sırası gelen oyuncu kendi kuyusundan aldığı taşları dağıtırken elinde taş kaldıysa, rakibinin bölgesindeki kuyulara da taş bırakmaya devam eder. Oyuncunun elindeki son taş, rakibinin bölgesinde denk geldiği kuyudaki taşların sayısını çft sayı yaparsa (2, 4, 6, 8 gibi) oyuncu bu kuyuda yer alan tüm taşların sahibi olur ve onları kendi hazinesine koyar. Hamle sırası rakibine geçer.

**3. TEMEL KURAL:** Oyuncu taşları dağıtırken elinde kalan son taş, yİne kendi bölgesinde yer alan boş bir kuyuya denk gelirse ve eğer boş kuyusunun karşısındaki kuyuda da rakibine ait taş varsa, hem rakibinin kuyusundaki taşları alır, hem de kendi boş kuyusuna bıraktığı taşı alıp hazinesine koyar. Hamle sırası rakibine geçer.

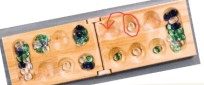

**4. TEMEL KURAL:** Oyunculardan herhangi birinin bölgesnde yer alan taşlar bittiğinde oyun seti biter. Oyunda kendi bölgesinde taşları ilk biten oyuncu, rakibinin bölgesinde bulunan tüm taşları da kazanır. Dolayısıyla, oyunun dinamiği son ana kadar hç düşmez. Mangala Oyunu, 5 set olarak oynanır. Oyunu kazanan oyuncu (1) puan, kaybeden (0) puan ve berabere bitten oyuncular yarım (0,5) puan alır.

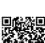

**Video İzle Öğren**

**Hazırlayan:**
**Ha** Yıl** Ku****
**Sa** El****

Kaynak https://www.mangala.com.tr/mangala-nasil-oynanir

**Fig 4. Infographic on Mangala designed on a free topic during the preliminary phase (Certain images have been replaced to comply with copyright requirements).**

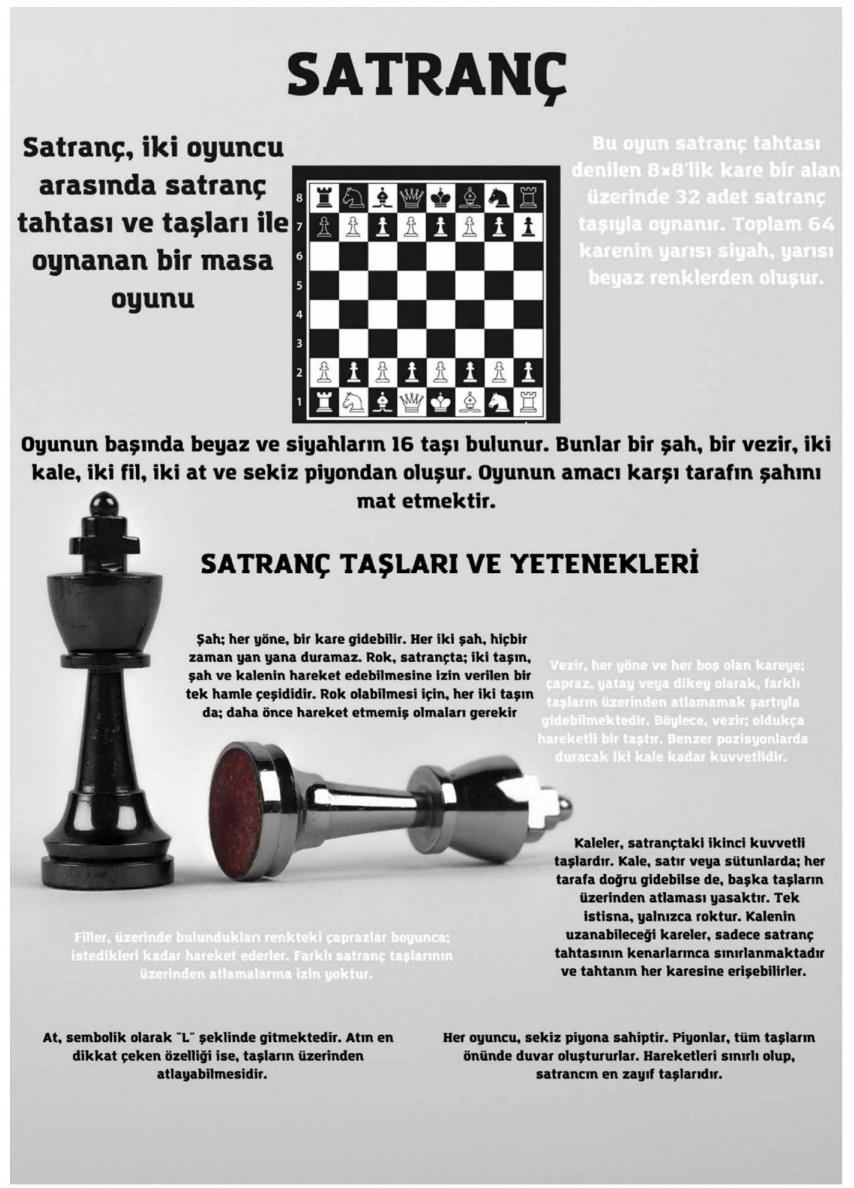

**Fig 5. Example of a chess-themed infographic designed on a free topic during the preliminary phase.**

research process of a narrator named Fatih. In addition, the prospective teacher attempted to support the content related to the channels through which bread is purchased and raising awareness about bread waste with visuals.

Fig 8 presents an infographic in which a prospective teacher attempted to present data on the remaining life expectancy of women and men according to age with the help of graphs. In addition, the prospective teacher attempted to show the rates of violence against women in countries with a pie chart and the employment rates of women and men over the years with a bar chart. The prospective teacher also included a human figure representing women and men in the infographic.

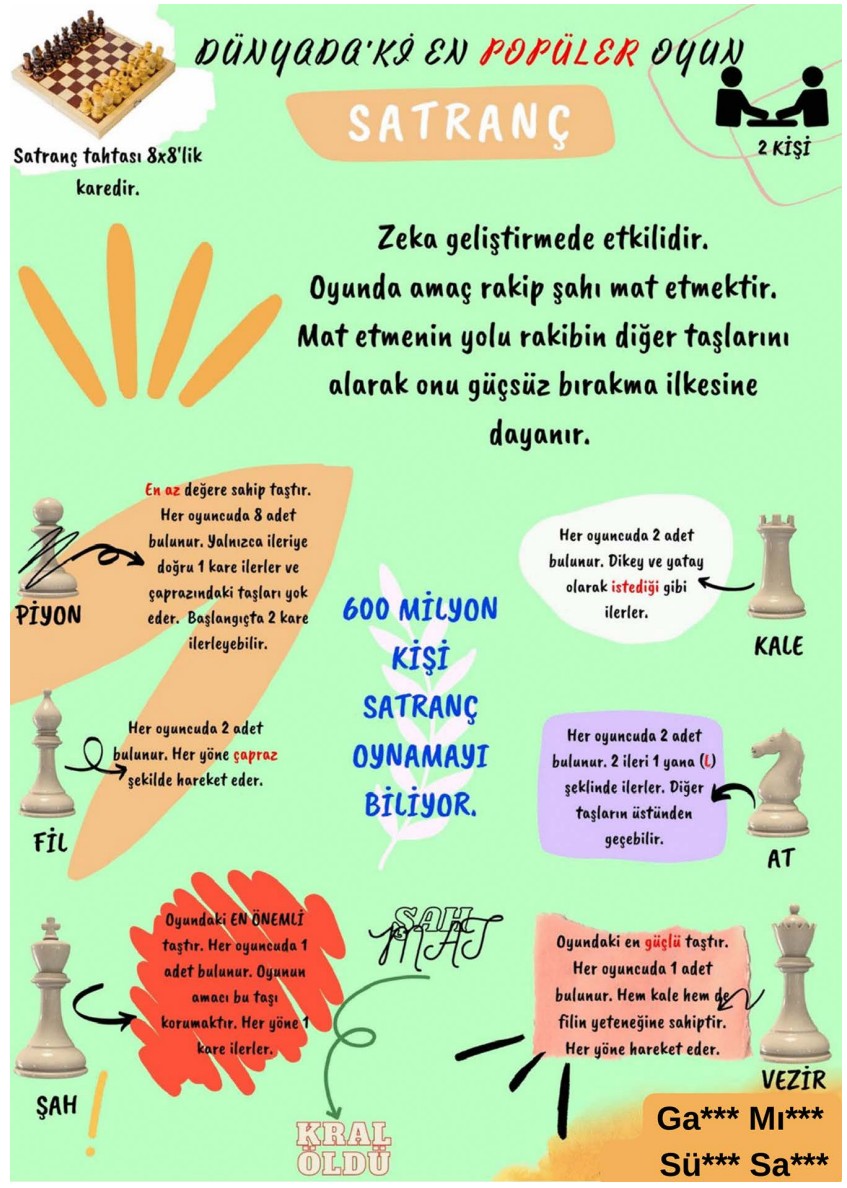

**Fig 6. Another example of a chess-themed infographic designed on a free topic during the preliminary phase.**

Since the Bread Waste and Women and Men infographics in Figs 7 and 8 bear only broad titles, the topic and type of information they present are not immediately clear at first glance. Moreover, infographics should engage the target audience, enable comprehension of the conveyed information, and aid retention. Although both infographics display data in brief texts and charts, the core message remains obscure. The Bread Waste infographic aims to collect data and visualize it with graphs to solve a problem introduced through a scenario that might be encountered in daily life. Despite its memorable and attention-grabbing storyline, the data are not presented effectively. In addition, the background layout hinders information tracking. Lack of color harmony and excessive ornamentation obscure the main message and prevent the infographic from being simple and effective. In the Women and Men infographic, improper use of color stemming from

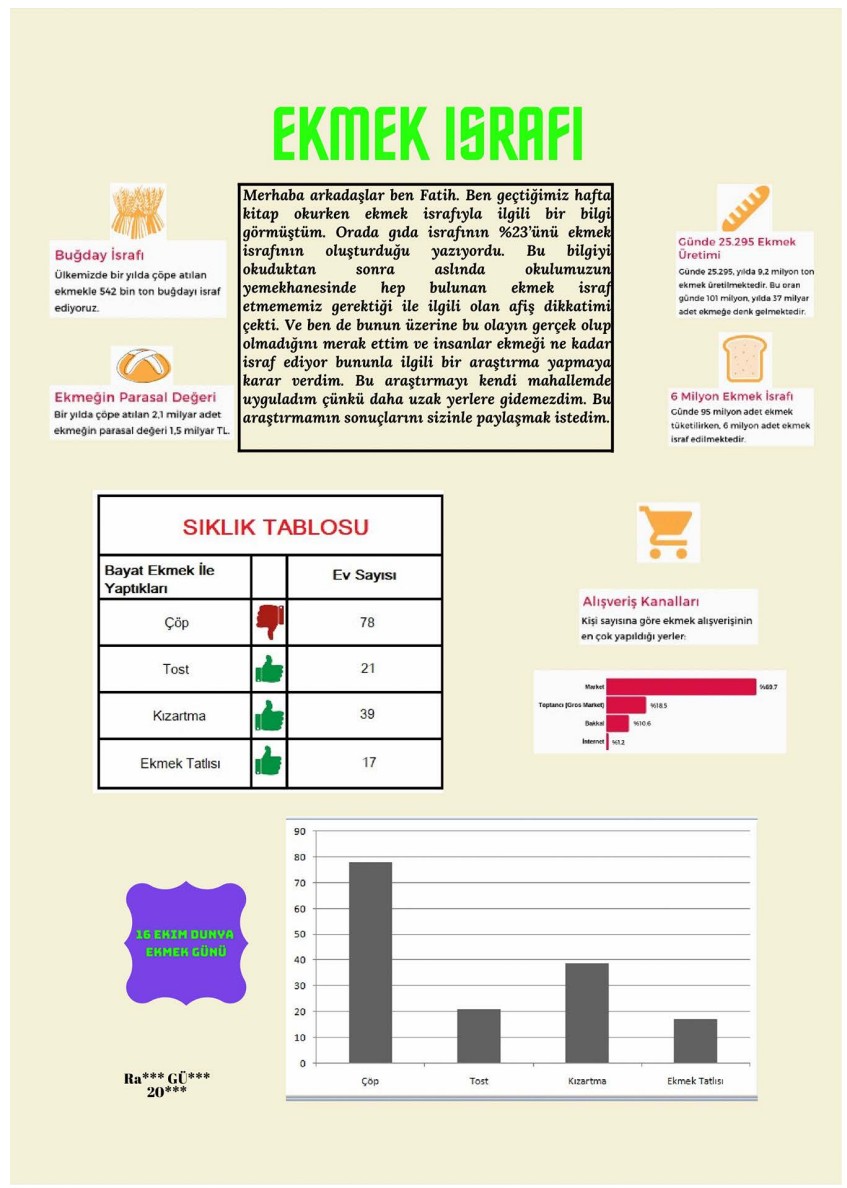

**Fig 7. Infographic entitled Bread Waste designed at the beginning of the main intervention.**

non-adherence to the principle of contrast makes the charted data difficult to read. The infographic combines unrelated data such as remaining life expectancy by age, employment rates, and violence against women across countries, leaving the main message unclear. Visual design principles are likewise not followed. Accordingly, when evaluated against the IDPR criteria, the Bread Waste infographic received a total score of 7, whereas the Women and Men infographic received a score of 4.

At the end of the main intervention, the prospective teachers designed infographics specific to the data processing learning domain. Examples of these infographics are presented in Figs 9–11.

Fig 9 presents an infographic in which a prospective teacher attempted to provide Türkiye's current population, population growth rate, and its change over the years through graphs. The prospective teacher also showed population ratios by

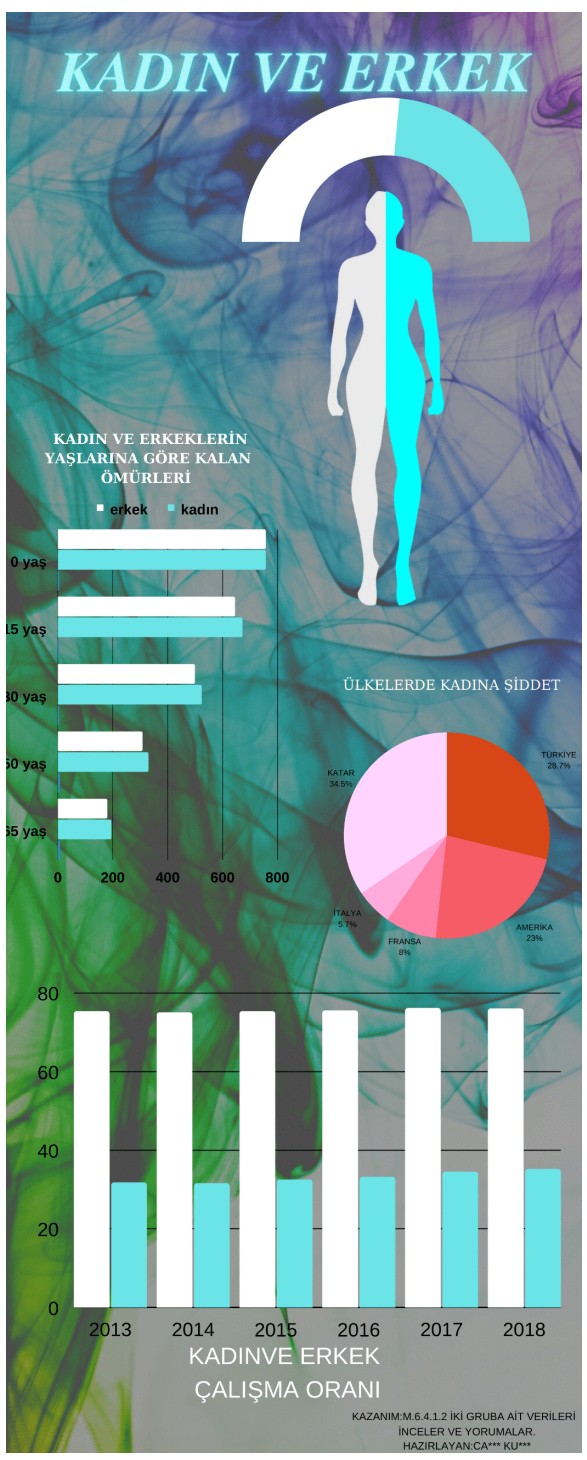

**Fig 8. Infographic entitled Women and Men designed at the beginning of the main intervention.**

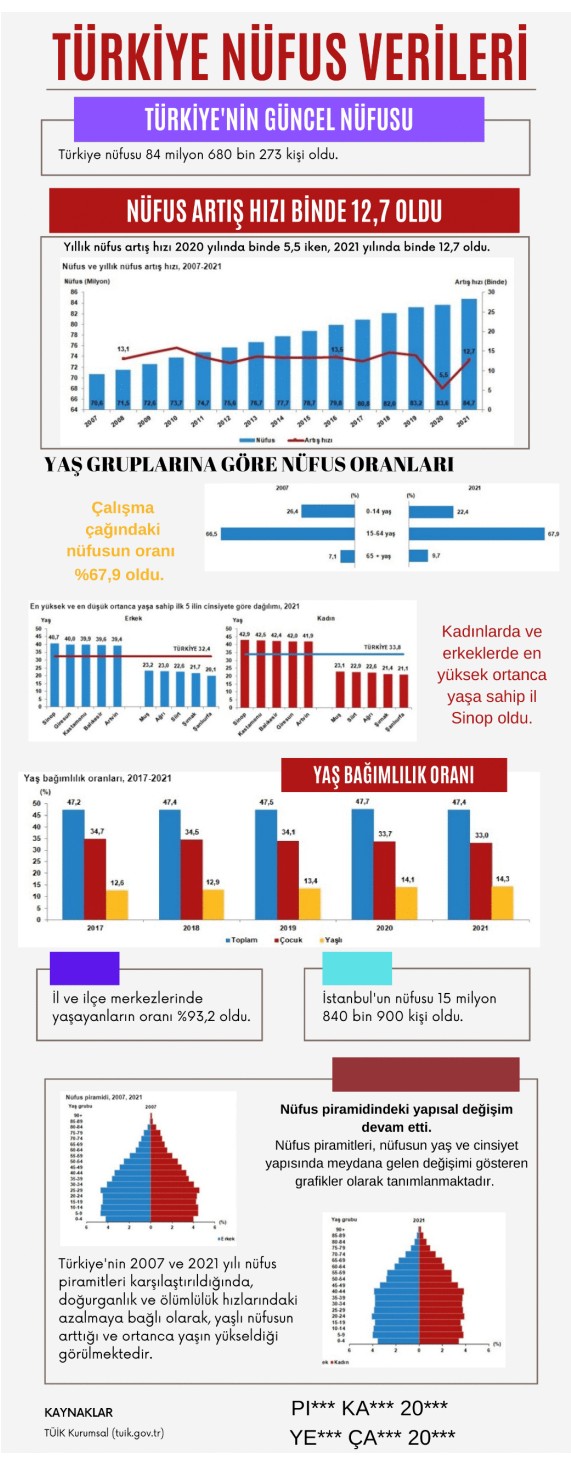

**Fig 9. Infographic entitled Türkiye's Population Data designed at the end of the main intervention.**

age groups, the provinces with the lowest and highest median ages, and age dependency ratios through various graphs. In addition, the prospective teacher included information regarding changes in the age and gender structure of the population through population pyramids.

Fig 10 presents an infographic in which a prospective teacher presented data related to the number of students taking the higher education entrance exam and the Basic Proficiency Test [Temel Yeterlilik Testi] (TYT) and Field Proficiency Test [Alan Yeterlilik Testi] (AYT). The prospective teacher attempted to convey achievement levels, mean, and standard deviation values for TYT and AYT with the help of pie and bar charts. The prospective teacher also visually expressed the proportions of successful and unsuccessful students through human figures represented in green, gray, and red colors.

In the Türkiye's Population Data infographic, simple information graphics were created to present the data, and complex information was displayed through a hierarchical arrangement of text and visuals. Explanatory text accompanies column and line charts, and the inclusion of references ensures that the data presented is current and accurate. Nevertheless, visual design principles are only partially observed. Colors are not employed effectively, and the placement of text and visuals on the background could be made more appealing. From a content perspective, however, it is evident that mathematical terms and expressions specific to the data processing domain are used correctly and that statements clarifying the data analysis methods are provided. The Mathematics Achievement infographic employs a general title and does not list any references. It depicts high-school students' performance in transitioning to higher education by means of various charts, yet the years to which the data pertain are not indicated. Although concepts such as mean, standard deviation, and arithmetic mean, which are specific to the data processing domain, are used, and shapes, column charts, and pie charts are employed for visualization, the infographic lacks a narrative supported by a meaningful title and a concise text that highlights the main message. Thus, this infographic can be said to align only partially with visual design principles. In this context, when assessed against the IDPR criteria, the Türkiye's Population Data infographic received a total score of 11, whereas the Mathematics Achievement infographic received a score of 9. An example created at the end of the main intervention is shown in Fig 11. Evaluated with the IDPR, Türkiye's Music-Listening Habits infographic scored 12.

Fig 11 presents an infographic in which a prospective teacher attempted to express what music means to individuals and the prominent characteristics according to different age groups in relation to music listening habits in Türkiye. The prospective teacher also showed the proportions of where and through which devices music is listened to with graphs and visuals. The prospective teacher indicated that pop music is the most preferred genre by presenting the distribution of music genres that people enjoy through a graph.

**Findings and interpretation on the change in prospective teachers' instructional explanations in the EEPrT and EEPoT according to the EER criteria. Findings and interpretation regarding the change in identifying errors and their causes:** Table 11 presents the frequencies of the prospective teachers' responses concerning the identification of middle school students' errors in the data processing learning domain and the underlying causes of those errors.

Table 11 indicates that the infographic design intervention positively influenced prospective teachers' instructional explanations when identifying middle school students' errors in the data processing learning domain and the underlying causes of those errors. The total number of responses falling into categories IE1 or IE2 in the post-test (23) decreased markedly compared with the pre-test (53). Conversely, the total number of responses in category IE4 rose substantially from the pre-test (147) to the post-test (192). A similar pattern emerged for the ICE categories. The combined total of responses in ICE1 and ICE2 dropped from 66 (pre-test) to 29 (post-test), whereas responses in ICE4 increased from 80 to 155. These findings suggest that the intervention enhanced the quality of prospective teachers' instructional explanations regarding the identification of student errors and their causes in the data processing learning domain. Below, one item from the pre-test (EEPrT) and one from the post-test (EEPoT) are provided, supported by excerpts from the teachers' instructional explanations.

In the first items labeled as *Muğla Vacation* in the EEPrT and *Charity Campaign* in the EEPoT, the middle school student's error lies in making intuitive selections and failing to recognize bias in the sample while disregarding the concept

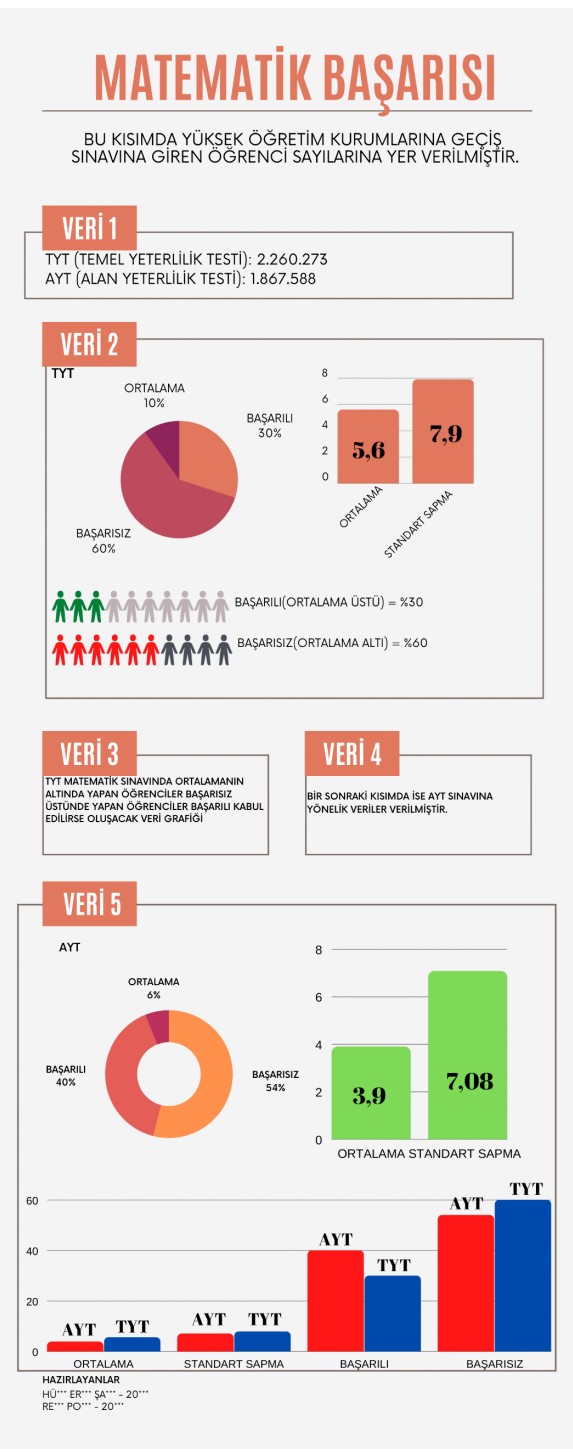

**Fig 10. Infographic entitled Mathematics Achievement designed at the end of the main intervention.**

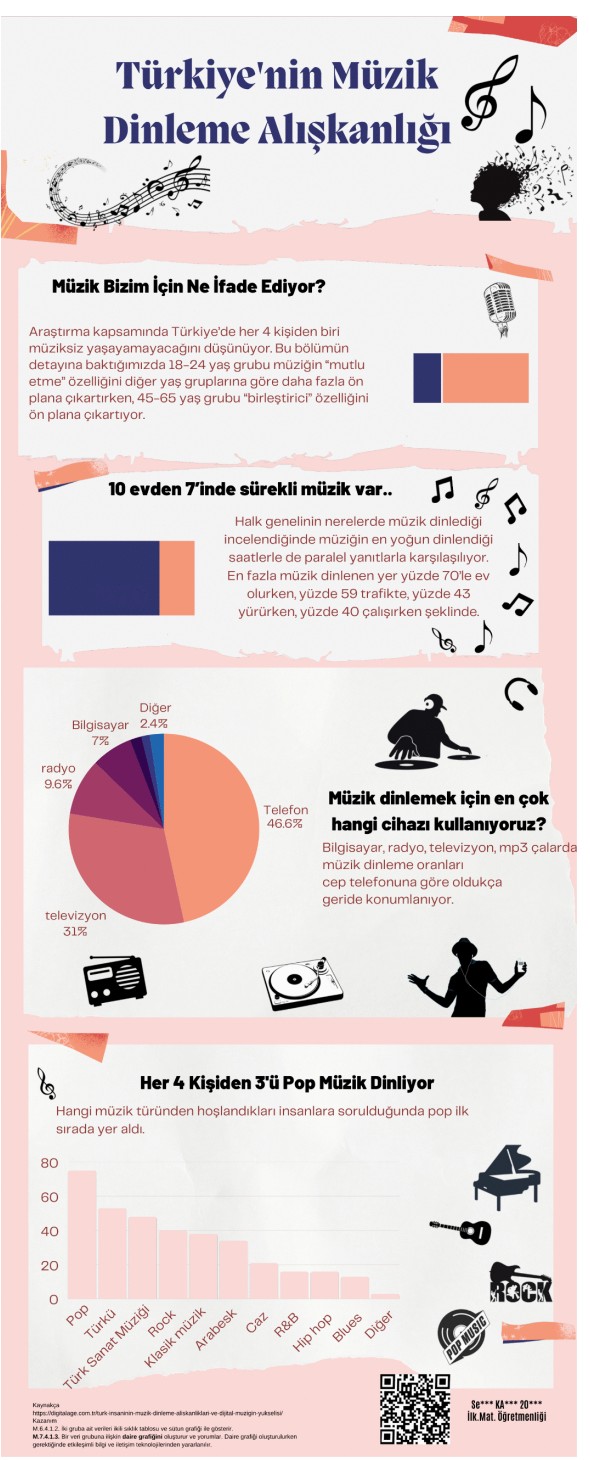

**Fig 11. Infographic entitled Türkiye's Music-Listening Habits designed at the end of the main intervention.**

**Table 11. Frequencies of prospective teachers' instructional explanations concerning the identification of errors and their causes.**

| Category | Sub Category | EEPrT | | | | | | EEPoT | | | | | |
|---|---|---|---|---|---|---|---|---|---|---|---|---|---|
| | | S1 f | S2 f | S3 f | S4 f | S5 f | Total f | S1 f | S2 f | S3 f | S4 f | S5 f | Total f |
| IE | IE1 | 0 | 5 | 4 | 17 | 3 | **29** | 5 | 4 | 1 | 4 | 1 | **15** |
| | IE2 | 1 | 1 | 7 | 15 | 0 | **24** | 4 | 0 | 2 | 2 | 0 | **8** |
| | IE3 | 9 | 4 | 4 | 4 | 4 | **25** | 3 | 0 | 0 | 5 | 2 | **10** |
| | IE4 | 35 | 35 | 30 | 9 | 38 | **147** | 33 | 41 | 42 | 34 | 42 | **192** |
| ICE | ICE1 | 0 | 9 | 5 | 21 | 3 | **38** | 5 | 4 | 2 | 5 | 1 | **17** |
| | ICE2 | 3 | 1 | 8 | 16 | 0 | **28** | 3 | 0 | 2 | 7 | 0 | **12** |
| | ICE3 | 12 | 23 | 13 | 6 | 15 | **69** | 12 | 4 | 10 | 7 | 8 | **41** |
| | ICE4 | 30 | 2 | 19 | 2 | 27 | **80** | 25 | 37 | 31 | 26 | 36 | **155** |

IE: Identifying the Error
IE1: Unable to identify the error/ no response
IE2: Incorrect identification of the error
IE3: Partially correct identification of the error
IE4: Correct identification of the error

ICE: Identifying the Cause of the Error
ICE1: Unable to identify cause of the error/ no response
ICE2: Incorrect cause of the error
ICE3: Partially correct cause of the error
ICE4: Correct cause of the error

of randomness. In the Muğla Vacation response, the student considered choosing a single store in Muğla sufficient for the research question and thus erred. The underlying reason is that the store is located in a tourist area, which the student deemed adequate. Similarly, in the Charity Campaign item, the student assumed that selecting only those in good financial standing would suffice, again making an error. By doing so, the student chose a sample that does not represent the entire population. The cause of the error is the student's focus on the well-off. Furthermore, the student assumed that items drawn "at random" from a bag would represent the whole population and ignored the fact that each selection must have an equal chance of being chosen. The student relied on intuition and personal opinion, which led to the mistake. For these questions, the prospective teachers are expected to detect the student's sampling bias, attend to sample size, choose representative samples, and pay heed to randomness. They should also explain that the errors may stem from students' conceptual deficiencies, particularly regarding percentage, probability, sampling, and randomness, and from their difficulty in making sense of these abstract notions. For example, in the *Muğla Vacation* item, PT1 misidentified the student's error and offered no explanation of its cause.

*PT1: "Choosing Muğla is correct, but I disagree that selecting people in the store is right. Asking people in Muğla does not make sense. Travel agency staff would know this better. I agree with the student."*

By contrast, in the *Charity Campaign* item on the post-test, PT41 correctly identified the error and provided an accurate explanation:

*PT41: "The student's answer is wrong. Students who are not well off should also be included in the sample because we want to gauge willingness to participate in the event. The student approached the problem with bias. That's likely why the error occurred. We want to determine the number of those unwilling to join the campaign as well. The student did not account for the negative case. The student may have assumed we only wanted information about participants. The student could not identify the sample well."*

**Findings and interpretation regarding the change in proposing corrective solutions:** Table 12 presents the frequencies of the prospective teachers' proposed solutions for addressing the errors made by middle school students in the data processing learning domain.

Table 12 indicates that the frequency of correct remedial suggestions offered by the prospective teachers for errors committed by middle school students in the relevant learning domain increased in the post-test compared with the pre-test. Specifically, the combined frequency for categories PS1 and PS2 fell from 86 (pre-test) to 53 (post-test), whereas the prospective teachers were able to propose many more fully correct solutions (82). The data thus reveal a clear shift from partially correct remedies toward entirely correct ones. The most striking detail in Table 12 is the rise in correct solution frequency from 21 to 82. These findings suggest that the infographic design instruction was highly effective in enabling prospective teachers to formulate accurate corrective strategies. An illustrative excerpt from PT1 that demonstrates the change and quality of the proposed remedies is presented below. Additional excerpts appear in Appendix 3.

PT1 (EEPrT – Football Match Game on Computer): *The student did not make an error. Therefore, I have no corrective suggestion.*

PT1 (EEPrT – Umut–Sevgi Middle Schools): *The student reached an incorrect result by merely adding the numbers without considering the mean of the data set. I would ask the student to verify the number of matches for the two schools and to reflect on whether simply summing the scores is fair if one school played fewer matches. The student needs to understand the concepts of mean and outliers to solve this problem. I would explain what mean, mode, median, and range signify and have the student engage in activities that demonstrate how these concepts are used in decision making. I would adopt a problem-based approach featuring contextualized problems to help the student internalize this knowledge.*

As exemplified by the excerpt from PT1, content analysis revealed that the solutions proposed by the prospective teachers after the infographic design instruction were more sophisticated than those suggested before the intervention. Whereas their pre-intervention suggestions tended to be superficial, the post-intervention recommendations aimed at fostering students' conceptual understanding and enabling learners to recognize and correct their own errors. Overall, the data indicate that the prospective teachers' post-intervention solution proposals gained in quality and foregrounded activities designed to remedy errors through diverse methods, techniques, and strategies. In general, the post-intervention solution proposals can be summarized as follows:

• Explaining the topic through examples drawn from everyday life,

• Providing opportunities for students to prepare and present their own examples,

• Requiring students to conduct research and collect data on the topic before class,

**Table 12. Frequencies of prospective teachers' corrective suggestions for students' errors.**

| Categories | Sub category | EEPrT | | | | | | EEPoT | | | | | |
|---|---|---|---|---|---|---|---|---|---|---|---|---|
| | | S1 f | S2 f | S3 f | S4 f | S5 f | Top. f | S1 f | S2 f | S3 f | S4 f | S5 f | Top. f |
| PS | PS1 | 11 | 15 | 9 | 29 | 8 | **72** | 19 | 5 | 5 | 11 | 6 | **46** |
| | PS2 | 0 | 1 | 6 | 7 | 0 | **14** | 4 | 0 | 1 | 2 | 0 | **7** |
| | PS3 | 32 | 26 | 25 | 7 | 28 | **118** | 13 | 26 | 23 | 11 | 17 | **90** |
| | PS4 | 2 | 3 | 5 | 2 | 9 | **21** | 9 | 14 | 16 | 21 | 22 | **82** |

PS: Proposing a Solution
PS1: No suggestion offered
PS2: Incorrect suggestion
PS3: Partially correct suggestion
PS4: Correct suggestion

- Posing guiding questions as the teacher and encouraging students to reason,

- Organizing activities that address missing concepts related to the topic,

- Employing problem-solving and discussion methods,

- Using computer-assisted mathematics teaching software such as GeoGebra,

- Discussing, through illustrative cases, the importance of random selection in representing a population,

- Debating classroom scenarios that challenge biased claims to foster relational understanding of sampling and population representation,

- Engaging in activities focused on data collection, analysis, and interpretation,

- Teaching the various types of graphs through hands-on activities,

- Working with real examples to practice graph interpretation.

Explanations of the responses to the items in the EEPrT and EEPoT, together with excerpts from the prospective teachers' instructional explanations, are provided in Appendix 4.

Table 13 presents the frequencies of the mathematical concepts and expressions that the prospective teachers used in their EEPrT and EEPoT responses specific to the data processing learning domain.

Table 13 shows that the number of mathematical concepts/expressions the prospective teachers used in the data processing learning domain increased from 73 in the pre-test to 121 in the post-test. Whereas 78 of the prospective teachers' pre-test responses contained no mathematical concept or expression, this number fell to 50 in the post-test. (These counts are based on the prospective teachers' answers to the EEPrT and EEPoT items that presented middle school students' erroneous solutions in data processing.) For example, in the Muğla Vacation item on the pre-test, PT38 wrote, *"I would ask whether the ten selected people actually went on holiday to Muğla and whether that is sufficient for the whole country. I don't know which method I should use to correct the error,"* thereby employing no mathematical concept or expression. Similarly, in the Charity Campaign item on the post-test, PT42 responded, *"Any amount of help matters. Even students who are not well off may wish to participate. I would tell my student this. I would add that even books they have read and no longer use could benefit pupils in a village school. I would highlight the importance of the book campaign. Students should decide for themselves whether to join,"* offering an off-topic, incorrect remedy without using any mathematical concept or expression. Table 13 also shows that the number of responses containing at least one mathematical concept or expression rose from 147 in the pre-test to 175 in the post-test. Overall, the total use of mathematical concepts or expressions increased markedly after the intervention, indicating that the infographic design instruction effectively

**Table 13. Frequencies of mathematical concepts/expressions used by prospective teachers in the EEPrT and EEPoT.**

| Category | Sub category | EEPrT | | | | | | EEPoT | | | | | |
|---|---|---|---|---|---|---|---|---|---|---|---|---|---|
| | | S1 f | S2 f | S3 f | S4 f | S5 f | Total f | S1 f | S2 f | S3 f | S4 f | S5 f | Total f |
| UMC | UMC1 | 17 | 7 | 15 | 32 | 7 | **78** | 20 | 7 | 5 | 11 | 7 | **50** |
| | UMC2 | 16 | 3 | 20 | 7 | 28 | **74** | 10 | 4 | 15 | 6 | 19 | **54** |
| | UMC3 | 12 | 35 | 10 | 6 | 10 | **73** | 15 | 33 | 25 | 28 | 20 | **121** |

UMC: Using Mathematical Concepts/Expressions
UMC1: No mathematical concept or expression used
UMC2: Only one mathematical concept or expression used
UMC3: At least two mathematical concepts or expressions used

expanded the prospective teachers' repertoire of mathematical language in the data processing learning domain. Table 14 lists the specific concepts and expressions used and shows how they changed by the end of the intervention.

When Table 14 is examined as a whole, it is evident that the overall frequency with which the prospective teachers used mathematical concepts or expressions specific to the data processing learning domain increased in the post-test (360) compared with the pre-test (253). The fact that the grand total in Table 14 exceeds that in Table 13 stems from the inclusion of instances in which at least two concepts or expressions were used. Table 14 shows that the concepts and expressions employed in both the pre- and post-tests pertain to the data processing domain. However, those appearing in the post-test are more closely related to the domain and include additional terms that support conceptual understanding. For example, unlike in the pre-test, the post-test features concepts and expressions such as *comparing data sets, data groups, equal chance of selection, data interpretation/processing/reading, measures of central tendency, discrete and continuous data, population, sampling, balanced distribution, homogeneous group, average value, and research method*. These findings suggest that the infographic design instruction not only increased the quantity of mathematical concepts and expressions that the prospective teachers used in the data processing domain but also enhanced their quality by reinforcing conceptual understanding.

**Table 14. Mathematical concepts or expressions used by prospective teachers in the EEPrT and EEPoT and their frequencies.**

| | EEPrT | | | | EEPoT | | | |
|---|---|---|---|---|---|---|---|---|
| Category | Mathematical Concept or Expression | f | Mathematical Concept or Expression | f | Mathematical Concept or Expression | f | Mathematical Concept or Expression | f |
| Subcategory | Mean | 33 | Ratio | 3 | Arithmetic mean | 29 | Percentage | 6 |
| | Arithmetic mean | 28 | Data collection | 3 | Line chart | 29 | Graph | 4 |
| | Percentage | 21 | Research question | 3 | Bar chart | 28 | Standard deviation | 4 |
| | Data | 16 | Representation | 3 | Ratio & proportion | 28 | Data processing | 3 |
| | Range | 16 | Pie sector | 3 | Graph interpretation | 25 | Representation | 3 |
| | Pie chart | 16 | Data set | 2 | Mean | 20 | Percentile | 3 |
| | Graph interpretation | 15 | Graph types | 2 | Modo | 20 | Evren | 2 |
| | Graph | 10 | Drawing graphs | 2 | Pie chart | 19 | Measures of central tendency | 2 |
| | Sample | 7 | Centre | 2 | Median | 12 | Table interpretation | 2 |
| | Percentile | 7 | Random distribution | 2 | Data comparison | 12 | Discrete/ continuous data | 2 |
| | Data analysis | 6 | Table | 2 | Data group | 12 | Sampling | 1 |
| | Bar chart | 6 | Fraction | 1 | Data | 10 | Fraction | 1 |
| | Line chart | 6 | Equal chance | 1 | Sample | 9 | Graph components | 1 |
| | Graph reading | 6 | Variation chart | 1 | Range | 9 | Percentage calculation | 1 |
| | Circle | 5 | Frequency | 1 | Survey | 8 | Contextual problem | 1 |
| | Standard deviation | 5 | Homogeneous distribution | 1 | Data collection | 7 | Balanced distribution | 1 |
| | Survey | 4 | Heterogeneous group | 1 | Data analysis | 7 | Random draw | 1 |
| | Statistics | 4 | Outliers | 1 | Angle measure | 8 | Homogeneous group | 1 |
| | Mode | 4 | Decimal representation | 1 | Data interpretation | 7 | Mean value | 1 |
| | Median | 4 | Data comparison | 1 | Random selection | 6 | Data reading | 1 |
| | Angle measure | 4 | Reliability | 1 | Independent / equal-probability selection | 6 | Statistics | 1 |
| | Graph title / elements | 3 | | | Drawing graphs | 6 | Research method | 1 |
| | **Sub Total** | **226** | | **37** | | **317** | | **43** |
| | **Grand Total** | **253** | | | | **360** | | |

## Limitations of the study

In this mixed-methods research, the qualitative analysis served to support the quantitative findings. Conversely, the absence of correlations among the independent variables suggests that the proposed model should be revised. The 16-week study was limited to a single research group of 45 third-year prospective elementary mathematics teachers at a public university in Türkiye. The discussion, conclusions, and recommendations that follow should therefore be considered within this scope.

## Discussion, conclusions, and recommendations

The analysis of the study's quantitative data shows that the intervention conducted through the infographic design instructional method had a statistically significant effect relative to the pre-intervention phase on improving prospective teachers' infographic design proficiency, their perceptions of infographic design self-efficacy, and their ability to evaluate middle school students' errors specific to the data processing learning domain.

Presenting the infographic design instructional process in a systematic manner (e.g., lecture via presentations, discussions through examples, preliminary practice, main intervention, and concurrent assessments) significantly enhanced the prospective teachers' abilities to visualize information, organize content, and create audience-appropriate designs. This result is largely supported by the descriptive analysis of the infographics they designed, evaluated against the IDPR criteria. The infographics designed after the intervention generally met the principles of infographic design and were somewhat higher in quality than those created before the intervention. Hence, the method employed can be considered effective in improving prospective teachers' infographic design proficiency. This finding is consistent with the results reported in studies by Kates et al. [36], Jaleniauskiene and Kasperiuniene [31], Nuhoğlu Kibar and Akkoyunlu [49], and Spector et al. [50]. During the design process, selecting information and arranging it hierarchically to best represent the topic or concept required prospective teachers to think critically and learn to condense material to fit a single-page template, thereby becoming active constructors of their own knowledge. Moreover, evaluating their design products through in-class discussions and feedback likely helped them improve their design proficiency. The relatively long 1-week intervention also afforded prospective teachers time for in-depth work, which may have contributed to producing higher-quality infographics than before the intervention. Elaldı and Çifçi [30] reported a similar pattern, finding the largest effect size for academic achievement in groups that used infographics for four to five weeks. Consequently, both quantitative and qualitative studies investigating how the length of the intervention influences prospective teachers' infographic design proficiency are recommended.

The infographic design instructional process increased prospective teachers' self-efficacy in designing infographics. Statistically significant differences with large effect sizes emerged across all subfactors of the IDSES. The greatest improvement occurred in the subfactor concerning competence in designing infographics appropriate to content and the target audience. This finding aligns with studies examining how designing and using infographics affect self-efficacy perceptions among teachers [17] and higher education students [25, 26]. Likewise, Fadzil's [37] results showed that prospective science teachers held positive views about using infographics they had created themselves. During that process, they participated actively and assumed responsibility for their own learning. The present findings mirror those results. Employing infographic design as an instructional method may have appealed to digitally immersed prospective teachers, sustaining their interest. Crucially, designing an infographic demands deep reflection and forces prospective teachers to consider how best to convey core messages and concepts to a target audience within limited space [37]. This feature likely prompted intensive engagement as prospective teachers mastered finer points of the topic. Their self-efficacy for infographic design appears to have grown accordingly.

The intervention of the infographic design instructional method was found to be effective in enhancing prospective teachers' abilities in evaluating middle school students' errors specific to the data processing learning domain. The greatest effects emerged in detecting the causes of errors and in overall error evaluation scores, comprising the identification

of the error and its cause, the provision of appropriate remedial suggestions, and the use of accurate domain-specific mathematical concepts and expressions. This result aligns with studies indicating that prospective teachers are generally successful in detecting student errors [20, 21]. After the intervention, the participants not only proposed suitable remedies but also offered qualitatively richer suggestions than before. Qualitative analyses revealed that, whereas pre-intervention solution proposals tended to be superficial, post-intervention suggestions addressed students' conceptual understanding and encouraged learners to recognize and correct their own errors. These findings are consistent with prior research showing that prospective teachers typically favor methods such as asking open-ended questions, providing examples, and employing visual materials when addressing student errors [17]. In addition, the present study found that participants recommended using computer-assisted mathematics teaching technologies (e.g., GeoGebra), assigning pre-class research tasks, and adopting inquiry, problem-solving, discussion, and problem-oriented strategies through daily life examples. Contrary to Cairo's [5] observation that some prospective teachers view student errors as personal failures, such an attitude did not emerge among the participants in this study. Qualitative evidence further showed that both the quantity and quality of the mathematical concepts and expressions used by the prospective teachers increased, thereby reinforcing conceptual understanding. Concepts that were absent in the pre-test, such as equal selection probability, data interpretation/processing/reading, measures of central tendency, discrete and continuous data, population, sampling, mean value, and research method, appeared in the post-test. Given that all participants had already completed courses such as Special Teaching Methods I–II, School Experience, Probability and Statistics, and Teaching Probability and Statistics, this improvement is particularly noteworthy. It may be attributed to the deep learning demands of infographic design, which compelled participants to engage in intensive study and critical thinking. Moreover, infographics likely facilitated the analysis and interpretation of complex data sets by presenting abstract data concepts visually, revealing relationships among data, and making information easily digestible [5, 6, 23]. Similar outcomes have been reported in studies using infographics with higher education students [28 –33].

In this study, a single-group pre-test–post-test quasi-experimental design was employed. The responses given by the prospective teachers to the open-ended questions were scored by the researchers through rubrics, these scores were transferred to the SPSS environment, and both quantitative and qualitative methods were used in the analysis of the data. This methodological diversity enhanced the reliability of the findings and enabled the results to be interpreted from multiple dimensions.

The most important limitation of the single-group pre-test–post-test quasi-experimental design is the difficulty in conclusively determining that the observed change resulted solely from the intervention. However, due to the nature of the design, certain potential threats to internal validity are present [71, 72]. In the study, the maturation factor may have affected the results to a small extent, as the prospective teachers could have shown natural cognitive and pedagogical development during the sixteen-week implementation period. In addition, testing effects and historical factors also make it difficult to conclusively attribute the observed positive effect solely to the intervention. Moreover, the prospective teachers' familiarity with the measurement process may have created a minor effect. However, the use of parallel tests in the study largely reduced the testing effect. This is because the prospective teachers did not gain an advantage in the post-test by recalling the exact questions they encountered in the pre-test; instead, they were presented with parallel test questions that were aligned with the same learning outcomes but expressed differently. This constitutes an important methodological precaution that minimizes the testing effect. On the other hand, other courses conducted at the university during the research period, additional resources used, or historical factors arising from the students' individual studies may also have influenced learning levels. This situation limits attributing the positive effect solely to the implemented instructional method and requires caution in interpreting the results. In terms of external validity, the fact that the study was limited to prospective teachers studying at a particular university restricts the generalizability of the findings to different student groups or contexts. The limitation of the setting and conditions to a single university context raises the question of whether similar results would be obtained in different institutions.

Furthermore, although the implementation being conducted by the same researcher–instructor provided an advantage in terms of consistency, it also raises the question of whether similar results would be obtained with different teachers or instructors. Finally, cultural and contextual factors limit the generalizability of the findings to different cultural settings. Despite these limitations, the absence of participant attrition, the consistent implementation of the instructional process, and the preparation of the measurement tools with parallel items were considered factors that reduced threats to internal validity [71, 72].

In summary, this study was conducted with prospective mathematics teachers studying at a particular university in Türkiye, within the context of the university's environment and conditions, and the intervention was carried out over sixteen weeks by a single instructor. This situation limits the generalizability of the research results to different groups, institutions, or cultural contexts. Therefore, in future studies, the inclusion of control or comparison groups, the replication of the intervention with different samples, and the implementation of long-term follow-up studies would strengthen both internal and external validity and contribute to grounding the findings in more robust causal inferences.

Another noteworthy finding is that no statistically significant correlations were detected among the scores from the IDSES, IDPR, and EER (EEPrT and EEPoT). This indicates that the observed gains in infographic design proficiency, self-efficacy, and error evaluation abilities occurred independently; that is, the instructional intervention affected each competency area separately. Consequently, the model proposed in this study delivers an important demonstration of how infographics can be employed in higher education courses and offers a concrete example of the outcomes such use can yield. This result aligns with the recommendations put forward by Jaleniauskiene and Kasperiuniene [31].

The proposed model seeks to give prospective teachers opportunities for in-depth engagement, thereby enhancing their subject matter mastery and self-efficacy in infographic design and enabling them both to create higher quality infographics and to improve their instructional explanations by evaluating student errors. Yet the lack of correlations among the dependent variables indicates that each was influenced independently by the intervention. Future work could examine prospective teachers' perspectives in depth, identify weaknesses or omissions in the model, and propose a more narrowly or broadly scoped version. Although the participants in the present study engaged in extensive work, research carried out explicitly within the principles of a deep learning approach could be considered. Since this investigation employed a single-group design, studies with larger samples that include both experimental and control groups could be conducted. Moreover, the study was confined to the data processing learning domain; similar research in other topics is recommended. Additional studies could explore how the infographic design instructional method affects prospective teachers' content knowledge and pedagogical content knowledge. Finally, integrating infographic design into higher education instructional processes is strongly recommended.

## Supporting information

**S1 Appendix. Infographic design self-efficacy scale (IDSES).**
(DOCX)

**S2 Appendix. Sample items from the error evaluation pre-test and post-test.**
(DOCX)

**S3 Appendix. Sample excerpts illustrating the change and improved quality of the prospective teachers' corrective suggestions.**
(DOCX)

**S4 Appendix. Detailed answers to the EEPrT / EEPoT items and sample instructional explanations provided by the prospective teachers.**
(DOCX)

## Acknowledgments

We would like to thank the prospective teachers and students who participated in this study for generously sharing their time and insights.

## Author contributions

**Formal analysis:** Neslihan Usta, Ali Özkaya, Gözdegül Arık Karamık.

**Investigation:** Neslihan Usta, Ali Özkaya, Gözdegül Arık Karamık.

**Methodology:** Neslihan Usta, Ali Özkaya, Gözdegül Arık Karamık.

**Project administration:** Ali Özkaya.

**Writing – original draft:** Neslihan Usta, Ali Özkaya, Gözdegül Arık Karamık, Yusuf Akın.

**Writing – review & editing:** Neslihan Usta, Ali Özkaya, Gözdegül Arık Karamık, Yusuf Akın.

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
