## [Decision Letter · Decision Letter 0]

28 Jan 2026

PONE-D-25-53363Examining the infographic design instructional process in terms of prospective mathematics teachers’ infographic design proficiency, self‑efficacy, and abilities in evaluating student errors: A model proposalPLOS One

Dear Dr. Özkaya,

Thank you for submitting your manuscript to PLOS ONE. After careful consideration, we feel that it has merit but does not fully meet PLOS ONE’s publication criteria as it currently stands. Therefore, we invite you to submit a revised version of the manuscript that addresses the points raised during the review process.

We look forward to receiving your revised manuscript.

Kind regards,

Jenna Scaramanga

Staff Editor

PLOS One

**Journal Requirements:**

1. When submitting your revision, we need you to address these additional requirements. Please ensure that your manuscript meets PLOS ONE's style requirements, including those for file naming. The PLOS ONE style templates can be found at https://journals.plos.org/plosone/s/file?id=wjVg/PLOSOne_formatting_sample_main_body.pdf and https://journals.plos.org/plosone/s/file?id=ba62/PLOSOne_formatting_sample_title_authors_affiliations.pdf 2. We note that you have indicated that there are restrictions to data sharing for this study. For studies involving human research participant data or other sensitive data, we encourage authors to share de-identified or anonymized data. However, when data cannot be publicly shared for ethical reasons, we allow authors to make their data sets available upon request. For information on unacceptable data access restrictions, please see http://journals.plos.org/plosone/s/data-availability#loc-unacceptable-data-access-restrictions.  Before we proceed with your manuscript, please address the following prompts: a) If there are ethical or legal restrictions on sharing a de-identified data set, please explain them in detail (e.g., data contain potentially identifying or sensitive patient information, data are owned by a third-party organization, etc.) and who has imposed them (e.g., a Research Ethics Committee or Institutional Review Board, etc.). Please also provide contact information for a data access committee, ethics committee, or other institutional body to which data requests may be sent. b) If there are no restrictions, please upload the minimal anonymized data set necessary to replicate your study findings to a stable, public repository and provide us with the relevant URLs, DOIs, or accession numbers. Please see http://www.bmj.com/content/340/bmj.c181.long for guidelines on how to de-identify and prepare clinical data for publication. For a list of recommended repositories, please see https://journals.plos.org/plosone/s/recommended-repositories. You also have the option of uploading the data as Supporting Information files, but we would recommend depositing data directly to a data repository if possible. Please update your Data Availability statement in the submission form accordingly. 3. Please include your full ethics statement in the ‘Methods’ section of your manuscript file. In your statement, please include the full name of the IRB or ethics committee who approved or waived your study, as well as whether or not you obtained informed written or verbal consent. If consent was waived for your study, please include this information in your statement as well. 4. Please remove your figures from within your manuscript file, leaving only the individual TIFF/EPS image files, uploaded separately. These will be automatically included in the reviewers’ PDF. 5. Please upload a new copy of Figures 4, 5 and 6, as the detail is not clear. Please follow the link for more information:  https://journals.plos.org/plosone/s/figures 6. We note that Figures 3, 4 and 7, in your submission contain copyrighted images. All PLOS content is published under the Creative Commons Attribution License (CC BY 4.0), which means that the manuscript, images, and Supporting Information files will be freely available online, and any third party is permitted to access, download, copy, distribute, and use these materials in any way, even commercially, with proper attribution. For more information, see our copyright guidelines: http://journals.plos.org/plosone/s/licenses-and-copyright. We require you to either present written permission from the copyright holder to publish these figures specifically under the CC BY 4.0 license, or remove the figures from your submission: a. You may seek permission from the original copyright holder of Figure(s) [#] to publish the content specifically under the CC BY 4.0 license.  We recommend that you contact the original copyright holder with the Content Permission Form (http://journals.plos.org/plosone/s/file?id=7c09/content-permission-form.pdf) and the following text:“I request permission for the open-access journal PLOS ONE to publish XXX under the Creative Commons Attribution License (CCAL) CC BY 4.0 (http://creativecommons.org/licenses/by/4.0/). Please be aware that this license allows unrestricted use and distribution, even commercially, by third parties. Please reply and provide explicit written permission to publish XXX under a CC BY license and complete the attached form.” Please upload the completed Content Permission Form or other proof of granted permissions as an "Other" file with your submission.  In the figure caption of the copyrighted figure, please include the following text: “Reprinted from [ref] under a CC BY license, with permission from [name of publisher], original copyright [original copyright year].” b. If you are unable to obtain permission from the original copyright holder to publish these figures under the CC BY 4.0 license or if the copyright holder’s requirements are incompatible with the CC BY 4.0 license, please either i) remove the figure or ii) supply a replacement figure that complies with the CC BY 4.0 license. Please check copyright information on all replacement figures and update the figure caption with source information. If applicable, please specify in the figure caption text when a figure is similar but not identical to the original image and is therefore for illustrative purposes only. 7. We note that Figure 7, in your submission contain map images which may be copyrighted. All PLOS content is published under the Creative Commons Attribution License (CC BY 4.0), which means that the manuscript, images, and Supporting Information files will be freely available online, and any third party is permitted to access, download, copy, distribute, and use these materials in any way, even commercially, with proper attribution. For these reasons, we cannot publish previously copyrighted maps or satellite images created using proprietary data, such as Google software (Google Maps, Street View, and Earth). For more information, see our copyright guidelines: http://journals.plos.org/plosone/s/licenses-and-copyright. We require you to either present written permission from the copyright holder to publish these figures specifically under the CC BY 4.0 license, or remove the figures from your submission: a. You may seek permission from the original copyright holder of Figure(s) [#] to publish the content specifically under the CC BY 4.0 license.   We recommend that you contact the original copyright holder with the Content Permission Form (http://journals.plos.org/plosone/s/file?id=7c09/content-permission-form.pdf) and the following text:“I request permission for the open-access journal PLOS ONE to publish XXX under the Creative Commons Attribution License (CCAL) CC BY 4.0 (http://creativecommons.org/licenses/by/4.0/). Please be aware that this license allows unrestricted use and distribution, even commercially, by third parties. Please reply and provide explicit written permission to publish XXX under a CC BY license and complete the attached form.” Please upload the completed Content Permission Form or other proof of granted permissions as an "Other" file with your submission. In the figure caption of the copyrighted figure, please include the following text: “Reprinted from [ref] under a CC BY license, with permission from [name of publisher], original copyright [original copyright year].” b. If you are unable to obtain permission from the original copyright holder to publish these figures under the CC BY 4.0 license or if the copyright holder’s requirements are incompatible with the CC BY 4.0 license, please either i) remove the figure or ii) supply a replacement figure that complies with the CC BY 4.0 license. Please check copyright information on all replacement figures and update the figure caption with source information. If applicable, please specify in the figure caption text when a figure is similar but not identical to the original image and is therefore for illustrative purposes only.The following resources for replacing copyrighted map figures may be helpful: USGS National Map Viewer (public domain): http://viewer.nationalmap.gov/viewer/The Gateway to Astronaut Photography of Earth (public domain): http://eol.jsc.nasa.gov/sseop/clickmap/Maps at the CIA (public domain): https://www.cia.gov/library/publications/the-world-factbook/index.html and https://www.cia.gov/library/publications/cia-maps-publications/index.htmlNASA Earth Observatory (public domain): http://earthobservatory.nasa.gov/Landsat: http://landsat.visibleearth.nasa.gov/USGS EROS (Earth Resources Observatory and Science (EROS) Center) (public domain): http://eros.usgs.gov/#Natural Earth (public domain): http://www.naturalearthdata.com/ 8. If the reviewer comments include a recommendation to cite specific previously published works, please review and evaluate these publications to determine whether they are relevant and should be cited. There is no requirement to cite these works unless the editor has indicated otherwise.

Reviewers' comments:

Reviewer's Responses to Questions

**Comments to the Author**

1. Is the manuscript technically sound, and do the data support the conclusions?

Reviewer #1: Yes

Reviewer #2: Yes

2. Has the statistical analysis been performed appropriately and rigorously? 

Reviewer #1: Yes

Reviewer #2: Yes

3. Have the authors made all data underlying the findings in their manuscript fully available?

The PLOS Data policy requires authors to make all data underlying the findings described in their manuscript fully available without restriction, with rare exception (please refer to the Data Availability Statement in the manuscript PDF file). The data should be provided as part of the manuscript or its supporting information, or deposited to a public repository. For example, in addition to summary statistics, the data points behind means, medians and variance measures should be available. If there are restrictions on publicly sharing data—e.g. participant privacy or use of data from a third party—those must be specified.requires authors to make all data underlying the findings described in their manuscript fully available without restriction, with rare exception (please refer to the Data Availability Statement in the manuscript PDF file). The data should be provided as part of the manuscript or its supporting information, or deposited to a public repository. For example, in addition to summary statistics, the data points behind means, medians and variance measures should be available. If there are restrictions on publicly sharing data—e.g. participant privacy or use of data from a third party—those must be specified.requires authors to make all data underlying the findings described in their manuscript fully available without restriction, with rare exception (please refer to the Data Availability Statement in the manuscript PDF file). The data should be provided as part of the manuscript or its supporting information, or deposited to a public repository. For example, in addition to summary statistics, the data points behind means, medians and variance measures should be available. If there are restrictions on publicly sharing data—e.g. participant privacy or use of data from a third party—those must be specified.requires authors to make all data underlying the findings described in their manuscript fully available without restriction, with rare exception (please refer to the Data Availability Statement in the manuscript PDF file). The data should be provided as part of the manuscript or its supporting information, or deposited to a public repository. For example, in addition to summary statistics, the data points behind means, medians and variance measures should be available. If there are restrictions on publicly sharing data—e.g. participant privacy or use of data from a third party—those must be specified.

Reviewer #1: Yes

Reviewer #2: No

4. Is the manuscript presented in an intelligible fashion and written in standard English?

Reviewer #1: Yes

Reviewer #2: Yes

5. Review Comments to the Author

**Reviewer #1:** After carefully reviewing the manuscript entitled:After carefully reviewing the manuscript entitled:After carefully reviewing the manuscript entitled:After carefully reviewing the manuscript entitled:

“Examining the infographic design instructional process in terms of prospective mathematics teachers’ infographic design proficiency, self‑efficacy, and abilities in evaluating student errors: A model proposal”

submitted by Neslihan Usta د, Ali Özkaya, Gözdegül Arık Karamık and Yusuf Akın,

I find that the study addresses an important and contemporary research topic that is well aligned with the aims and scope of the Journal. The manuscript demonstrates originality, clearly defined objectives, and an appropriate methodological framework.

The authors employed a research methodology suitable for the nature of the study, and the findings show logical consistency with the research questions and objectives. The results are adequately analyzed and supported by relevant and properly cited scholarly sources. Furthermore, the manuscript is well organized, clearly presented, and written in sound academic English.

Based on the above considerations, I recommend acceptance of the manuscript for publication

yours sincerely,

**Reviewer #2:** The manuscript presents a well-structured and methodologically sound investigation into the effects of an infographic design–based instructional process on prospective mathematics teachers’ design proficiency, self-efficacy, and abilities to evaluate student errors. The study addresses a relevant and timely topic in mathematics education and educational technology, and it aligns well with the scope of PLOS ONE.The manuscript presents a well-structured and methodologically sound investigation into the effects of an infographic design–based instructional process on prospective mathematics teachers’ design proficiency, self-efficacy, and abilities to evaluate student errors. The study addresses a relevant and timely topic in mathematics education and educational technology, and it aligns well with the scope of PLOS ONE.The manuscript presents a well-structured and methodologically sound investigation into the effects of an infographic design–based instructional process on prospective mathematics teachers’ design proficiency, self-efficacy, and abilities to evaluate student errors. The study addresses a relevant and timely topic in mathematics education and educational technology, and it aligns well with the scope of PLOS ONE.The manuscript presents a well-structured and methodologically sound investigation into the effects of an infographic design–based instructional process on prospective mathematics teachers’ design proficiency, self-efficacy, and abilities to evaluate student errors. The study addresses a relevant and timely topic in mathematics education and educational technology, and it aligns well with the scope of PLOS ONE.

The research design is clearly described, and the instructional intervention is implemented over an adequate duration with transparent documentation of procedures. The use of a mixed-methods approach, combining quantitative analyses with qualitative content analysis of infographics and error-evaluation responses, strengthens the overall rigor of the study. The selected instruments are grounded in prior literature, and the authors have provided appropriate evidence of reliability and inter-rater agreement, which enhances confidence in the measurements.

The statistical analyses are generally appropriate and rigorously applied. The authors correctly selected parametric and non-parametric tests based on data distribution, reported effect sizes, and presented results in a clear and interpretable manner. The findings consistently support the stated conclusions regarding improvements in infographic design proficiency, self-efficacy, and error-evaluation abilities following the intervention.

However, there are several points that should be addressed to further strengthen the manuscript. First, the reliance on a single-group pre-test–post-test quasi-experimental design limits the ability to draw strong causal inferences. Although the authors acknowledge this limitation, it would be beneficial to discuss more explicitly how potential threats to internal validity (e.g., maturation, testing effects) may have influenced the results and how future studies could address these issues through control or comparison groups.

Second, while the Data Availability Statement explains restrictions related to participant consent and confidentiality, the manuscript does not provide access to anonymized datasets or example data files underlying the quantitative analyses. This only partially meets PLOS ONE’s data-sharing requirements. The authors are encouraged to clarify whether de-identified or aggregated datasets, scoring rubrics, or sample coded data can be shared as Supporting Information to enhance transparency and reproducibility.

Third, although the manuscript is generally well written and intelligible, minor language and stylistic issues remain, including occasional long or repetitive sentences and minor grammatical inconsistencies. Careful proofreading is recommended to improve clarity and conciseness, particularly in the Results and Discussion sections.

From an ethical and publication ethics perspective, the study appears to comply with relevant standards. Ethical approval is clearly documented, informed consent was obtained, and no competing interests or dual publication concerns are evident.

Overall, the manuscript makes a meaningful contribution to the literature on infographic-based instruction in teacher education. With clearer acknowledgment of design limitations, improved data availability where possible, and minor language refinements, the study would be further strengthened and suitable for publication.

6. PLOS authors have the option to publish the peer review history of their article (what does this mean?). If published, this will include your full peer review and any attached files.). If published, this will include your full peer review and any attached files.). If published, this will include your full peer review and any attached files.). If published, this will include your full peer review and any attached files.

...

Reviewer #1: **Yes:** Lina Fouad JawadLina Fouad JawadLina Fouad JawadLina Fouad Jawad

Reviewer #2: **Yes:** Muralidhar KurniMuralidhar KurniMuralidhar KurniMuralidhar Kurni

---

## [Author Response · Author response to Decision Letter 1]

12 Feb 2026

The detailed responses to all reviewer comments are provided in the ‘Response to Reviewers’ document.

---

## [Editor Report · Decision Letter 1]

26 Mar 2026

Examining the infographic design instructional process in terms of prospective mathematics teachers’ infographic design proficiency, self‑efficacy, and abilities in evaluating student errors: A model proposal

PONE-D-25-53363R1

Dear Dr. Özkaya,

We’re pleased to inform you that your manuscript has been judged scientifically suitable for publication and will be formally accepted for publication once it meets all outstanding technical requirements.

Kind regards,

Sonsoles López-Pernas

Academic Editor

PLOS One

Additional Editor Comments (optional):

The authors have successfully addressed all minor comments

---

## [Editor Report · Acceptance letter]

PONE-D-25-53363R1

PLOS One

Dear Dr. Özkaya,

I'm pleased to inform you that your manuscript has been deemed suitable for publication in PLOS One. Congratulations! Your manuscript is now being handed over to our production team.

Kind regards,

on behalf of

Dr Sonsoles López-Pernas

Academic Editor

PLOS One